# TAMING DIFFUSION TRANSFORMER FOR EFFICIENT MOBILE VIDEO GENERATION IN SECONDS

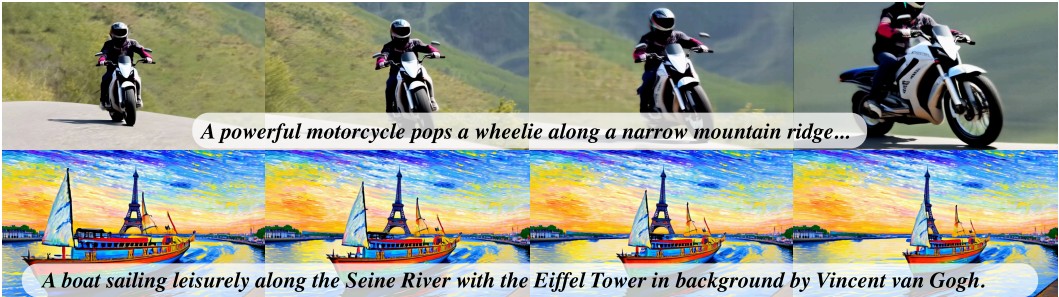

Figure 1: Videos generated by our efficient Diffusion Transformer.

## ABSTRACT

Diffusion Transformers (DiT) have shown strong performance in video generation tasks, but their high computational cost makes them impractical for resource-constrained devices like smartphones, and practical on-device generation is even more challenging. In this work, we propose a series of novel optimizations to significantly accelerate video generation and enable practical deployment on mobile platforms. First, we employ a highly compressed variational autoencoder (VAE) to reduce the dimensionality of the input data without sacrificing visual quality. Second, we introduce a KD-guided, sensitivity-aware tri-level pruning strategy to shrink the model size to suit mobile platforms while preserving critical performance characteristics. Third, we develop an adversarial step distillation technique tailored for DiT, which allows us to reduce the number of inference steps to four. Combined, these optimizations enable our model to achieve approximately 15 frames per second (FPS) generation speed on an iPhone 16 Pro Max, demonstrating the feasibility of efficient, high-quality video generation on mobile devices.

## 1 INTRODUCTION

The rapid advancement of generative models (Liu et al., 2022; Esser et al., 2024; Rombach et al., 2022) has led to significant breakthroughs in video generation (Blattmann et al., 2023; OpenAI, 2023; Zheng et al., 2024; Wan et al., 2025; Yang et al., 2024b; Team, 2024; Kong et al., 2024), with Diffusion Transformers emerging as one of the most effective architectures for producing temporally coherent and visually compelling video content. These models leverage the strengths of diffusion processes (Rombach et al., 2022; Ho et al., 2020; Liu et al., 2022) for stepwise refinement and transformer-based attention for capturing long-range dependencies across frames, making them particularly suitable for generating complex, high-fidelity video sequences. As such, they have become a cornerstone in state-of-the-art video synthesis pipelines.

Despite their impressive generative capabilities, Diffusion Transformers suffer from substantial computational overhead, especially when applied to high-resolution video generation. While this brings significant quality improvements, the computation and memory consumption of the 3D full attention (Yang et al., 2024b; Kong et al., 2024) scale quadratically with respect to total tokens ($t \times H \times W$). This limitation poses a critical challenge for deploying these models in interactive settings, particularly on mobile devices with limited processing power and energy budgets. Existing efforts to optimize diffusion-based models, such as step reduction (Zhang et al., 2024; Li et al., 2023; Yin et al., 2024a), efficient backbone (Wu et al., 2025b; Yahia et al., 2024; Zhao et al., 2024), are mainly focused on UNet-based denoisers, which are naturally less expressive. Very few

work (HaCohen et al., 2024b) investigates the efficiency of DiT, and often suffers from perceptual quality or temporal consistency loss. Furthermore, most current acceleration methods are designed for desktop (HaCohen et al., 2024b; Wan et al., 2025), and do not translate well to edge devices.

In this work, we present a comprehensive optimization pipeline tailored specifically to accelerate video diffusion transformers for mobile deployment. Our approach combines four key strategies.

Table 1: Our model is the first DiT-based mobile video generator. Generation speed is reported as FPS. See Sec. 5 for details.

| Model | Params (B) | VBench | A100 | iPhone |
|---|---|---|---|---|
| Wan2.1 | 1.3 | 83.33 | 0.2 | ✗ |
| LTX | 1.8 | 80.00 | 6.1 | ✗ |
| Ours-Server | 2.0 | 83.09 | 6.4 | ✗ |
| Ours-Mobile | 0.9 | 81.45 | 151.3 | ∼15 |

**(A) High-Compression Video Variational Autoencoder (VAE).** First, we investigate the compression rate of the VAE. High compression video VAE can significantly reduce the number of tokens in the latent representation, thus speeding up DiT inference. However, VAEs with an aggressive compression ratio often suffer from a loss of reconstruction quality, which likely leads to a loss in diffusion model generation quality. The trade-off between compression ratio and diffusion quality remains underexplored for on-device video generation. In this work, we create a series of video VAEs with different compression ratios and compare the speed gain versus generation quality loss. We have several findings: (i) The reconstruction and diffusion generation quality corresponds well with the compression ratio. (ii) The speedup from the higher VAE compression ratio is significant. (iii) Though slightly degraded, we can still find a sweet point that balances speed and quality.

**(B) Efficient Mobile DiT.** Second, we find that directly training a lightweight DiT designated for mobile is challenging. Instead, we start from a larger pre-trained supernet and propose a sensitivity-aware tri-level pruning with a KD-Guided framework that selectively removes less critical components of the model based on their contribution to both runtime and output quality. This pruning reduces the number of DiT blocks, feed-forward features, and attention heads. The final architecture has 915M parameters and can be easily deployed on a modern device such as an iPhone 16 Pro Max. Further, we improve the pruned model performance by aligning features of the pruned network and the supernet through knowledge distillation.

**(C) Adversarial Step Distillation.** Third, we design a new discriminator head tailored for adversarial step-distillation on DiTs that achieves full-step quality with only a few sampling steps. Prior adversarial distillation methods for video diffusion models mainly focus on UNet backbones (Yahia et al., 2024; Zhang et al., 2024) and less challenging image-to-video tasks (Blattmann et al., 2023) and do not transfer directly to DiTs. Our discriminator design inherits the first $K$ frozen pretrained generator blocks as a time-conditioned feature parser, and adds learnable with self-attention and cross-attention to fully capture conditions. This design enables *four-step* generation without classifier-free guidance (CFG), yielding $20\times$ faster inference than a typical 40-step CFG recipe.

**(D) Operator Optimization for Efficient Inference.** Finally, we identify a memory bottleneck in the linear layer of DiTs (*i.e.* feed-forward network), where the limited bandwidth prevents operators from approaching their theoretical speed on device. To address this, we introduce a tiled GEMM strategy that alleviates the memory bottleneck without requiring kernel-level modifications. The design achieves over 50% speedup on targeted linear layers and $\sim$10% acceleration for DiT inference.

With these optimizations, our model can generate high-quality video at over $15$ frames per second (FPS) on an iPhone 16 Pro Max using only four denoising steps. Extensive experiments demonstrate that our method maintains strong visual fidelity and temporal consistency, closely matching the outputs of full-resolution, unpruned models. For the first time, our work advances the state-of-the-art for on-device efficient video generation by making practical diffusion-based video synthesis feasible on consumer-grade mobile hardware. Our contributions can be summarized as follows,

- We are the first to systematically investigate the trade-off between latent compression ratio, generation quality, and speed for on-device video generation. We find that for diffusion transformer, a $8 \times 32 \times 32$ VAE achieves a good trade-off between generation speed and quality.

- To obtain an efficient DiT backbone, training a smaller network from scratch gives inferior results. Instead, we start from a large pre-trained super network and apply distillation-guided, sensitivity-aware pruning, yielding a compact network with optimized depth and width.

- For adversarial step-distillation, we propose a new discriminator design tailored for DiTs which outperforms prior methods by a large margin. We achieve $4$-step inference without CFG.

- We identify the memory bottleneck in the feed-forward layers of DiTs on device performance and introduce a tiled GEMM strategy that alleviates this issue, enabling more efficient and hardware friendly on-device inference.

## 2 RELATED WORK

**Video Diffusion Models.** Recent years have seen rapid progress in video generation models (Wan et al., 2025; OpenAI, 2023; Lin et al., 2024; Yang et al., 2024b; Kong et al., 2024; Team, 2024; Kuaishou, 2024; Ma et al., 2025). Most advances focus on large diffusion models that iteratively denoise Gaussian noise into realistic videos, conditioned on text or images. These approaches include pixel-space models (Menapace et al., 2024; Ho et al., 2022) and latent-space models (Wan et al., 2025; Yang et al., 2024b). While such systems (OpenAI, 2023; Zheng et al., 2024; Menapace et al., 2024; Polyak et al., 2024; HaCohen et al., 2024b; Yang et al., 2024b) generate high-quality videos, their resource demands make them unsuitable for on-device use.

**On-Device Models.** In contrast, only limited work targets on-device video generation (Wu et al., 2025b; Kim et al., 2025). The Wan2.1 family (Wan et al., 2025) includes a 1.3B T2V model, but its low VAE compression yields too many latent tokens for deployment. LTX-Video (HaCohen et al., 2024a) applies a high-compression VAE and runs in real time on GPUs, yet its 1.9B parameters remain prohibitive for mobile devices. SnapGen-V (Wu et al., 2025b) adopts a lightweight UNet but sacrifices visual fidelity. Mobile Video Diffusion (Yahia et al., 2024) reduces Stable Video Diffusion (Blattmann et al., 2023) by pruning channels and blocks. On-device Sora (Kim et al., 2025) achieves low-resolution video generation on iPhones via temporal token merging and concurrent block loading to handle memory limits.

**Step Distillation.** Diffusion models (Esser et al., 2024; Podell et al., 2023; Hoogeboom et al., 2023) require many denoising steps, each involving a full network pass, which creates latency. Reducing the number of steps directly improves efficiency. Numerous methods address this in text-to-image tasks (Yin et al., 2024b;a; Yang et al., 2024a; Wang et al., 2024b; Kim et al., 2024; Mei et al., 2024; Dao et al., 2025), representative work including progressive distillation (Salimans & Ho, 2022; Li et al., 2023), consistency models (Song et al., 2023; Song & Dhariwal, 2023), adversarial training (Xu et al., 2024; Sauer et al., 2023b; 2024), shortcut models (Frans et al., 2024), and mean flow (Geng et al., 2025). For video, Zhang et al. (2024); Wu et al. (2025b) achieves few-step generation Blattmann et al. (2023) with adversarial training specially-designed spatio-temporal discriminator.

## 3 PRELIMINARIES

Following popular practices of latent diffusion (Zheng et al., 2024), we employ a video autoencoder to encode video data $\mathbf{X} \in \mathbb{R}^{3 \times T \times H \times W}$ into a compressed latent space $\mathbf{x} \in \mathbb{R}^{c \times t \times h \times w}$, where $T$ is the number of temporal frames, $H$ and $W$ are the spatial resolutions, and $c$ is the latent channels. The VAE compression ratio is thus $\frac{T}{t} \times \frac{H}{h} \times \frac{W}{w}$, e.g., $4 \times 8 \times 8$ (Yang et al., 2024b; Kong et al., 2024; Wan et al., 2025) and $8 \times 16 \times 16$ (Agarwal et al., 2025) VAEs. The objective of the DiT generator is to generate $\mathbf{x}$ under certain guidance (i.e., text prompt).

We employ Rectified Flow (Wang et al., 2024b) to train our latent DiT model. According to the flow-matching-based diffusion process, given a clean video latent $\mathbf{x}_0 = \mathbf{x}$, the intermediate noisy state $\mathbf{x}_t$ at a timestep $t$ is:

$$\mathbf{x}_t = (1 - t) \mathbf{x}_0 + t\epsilon, \text{ where } \epsilon \sim \mathcal{N}(0, I), \tag{1}$$

which is a linear interpolation between the data distribution and a standard normal distribution. The model aims to learn a vector field $v_\theta(t, \mathbf{x}_t)$ using the Conditional Flow Matching objective, *i.e.*,

$$\mathcal{L}_{\text{fm}} = \mathbb{E}_{t, \epsilon, \mathbf{x}_0} \|v_\theta(t, \mathbf{x}_t) - (\epsilon - \mathbf{x}_0)\|_2^2. \tag{2}$$

## 4 METHOD

We optimize Diffusion Transformer (DiTs) for efficient on-device video generation from four perspectives: **(a) High compress VAE:** we employ a high-compression autoencoder, as the computational complexity of transformer scales quadratically with token length. Reducing the number of token

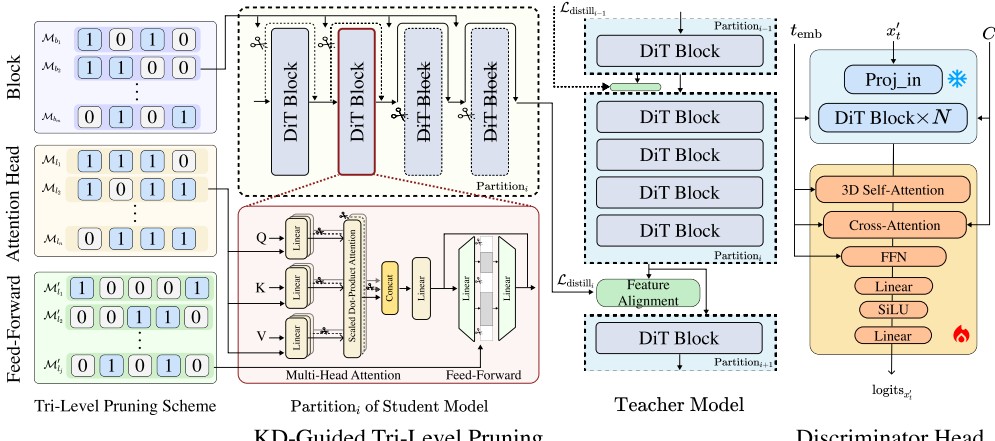

Figure 2: **Overview of proposed KD-Guided Tri-Level Pruning and new discriminator head.** The tri-level pruning scheme operates across three levels of granularity, the block, attention-head, and feed-forward network dimension, ranging from coarse to fine. This design enables flexible, efficient, and stable model compression. Additionally, the proposed discriminator adopts standard DiT blocks with a MLP classifier head, improved condition alignment for adversarial training.

decreases computation while enabling video generation at higher resolution and longer duration. **(b) Efficient DiT architecture:** we design an efficient DiT using a KD-Guided Tri-Level pruning method. The method balances model fidelity to the baseline with the hardware constraint of the target device. **(c) Step-distillation:** we adopt adversarial fine-tuning for step distillation with a new discriminator head design, reducing the number of sampling steps and achieving up to $20\times$ acceleration during inference. **(d) Operator Optimization:** we identify the memory bottlenecks in DiT feed-forward layers and introduce a tiled GEMM strategies, alleviating bandwidth limitations and enabling efficient on-device inference without requiring kernel/compiler modification.

## 4.1 SCALING LATENT COMPRESSION RATIO

DiT demonstrates superior generation capabilities when attending on full token length ($thw$), however, it is also notorious for its quadratic computational cost. The key idea of the latent diffusion model is to construct a compressed latent space and reduce the generation cost. As a result, a straightforward idea to accelerate DiT is to further increase the VAE compression ratio. State-of-the-art models (CogVideoX (Yang et al., 2024b),Hunyuan(Kong et al., 2024),Wan (Wan et al., 2025)) employ a $4 \times 8 \times 8$ VAE combined with a $1 \times 2 \times 2$ patchify module, which comprises a $4 \times 16 \times 16$ total compression rate, while the recent OpenSora-2 (Zheng et al., 2024) adopts a $4 \times 32 \times 32$ VAE, and LTX (HaCohen et al., 2024b) adopts an $8 \times 32 \times 32$ VAE to reduce the dimensionality of the latent features input to the DiT and results in faster generation speed. However, there has been limited research on how the VAE compression ratio affects the quality and speed of video generation. Upon aggressive compression, it becomes more challenging for the VAE decoder to fully reconstruct the details, which may result in quality loss. In this work, we perform a comprehensive study on the scaling of the VAE compression ratio. We follow HaCohen et al. (2024b); Wu et al. (2025a) and construct video VAEs with various compression ratios from $4 \times 16 \times 16$ to $8 \times 64 \times 64$. We build the VAE with 3D convolutions to better handle video modality, and use a fixed latent channel number for all variants. We train the same DiT network under each latent space and benchmark the generation speed and quality. Results and discussions are in Sec. 5.3 and Appendix F.

## 4.2 EFFICIENT DIT ARCHITECTURE VIA KD-GUIDED TRI-LEVEL PRUNING

Despite operating in a highly compressed latent space, the size of the Diffusion Transformer (DiT) remains a critical factor in edge generation scenarios (Wu et al., 2025b), where mobile devices are constrained by limited memory, power, and computational resources. Training a compact DiT that still achieves high-quality generation is a non-trivial challenge. First, DiT models generally exhibit

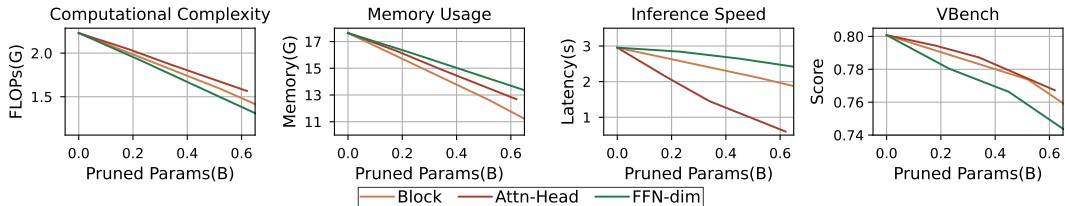

Figure 3: Sensitivity Analysis of DiT Components. The sensitivity analysis is conducted by progressively pruning DiT blocks, attention-heads and feed-forward network (FFN) dimension. For each setting, we benchmark FLOPs, memory usage, inference speed, and VBench score to assess the impact of each component on model efficiency and performance.

strong generation capabilities only when scaled to a sufficiently large capacity. Moreover, designing an effective small-scale DiT is difficult due to the high-dimensional design space—including network depth (number of transformer blocks), width (channel size), and attention head count.

A promising alternative is to begin with a well-trained large model and prune it to meet resource constraints. Prior work such as TinyFusion (Fang et al., 2024) explored this approach via block-wise pruning using a learnable layer mask, effectively constructing a shallower DiT. However, as the properties demonstrated in Figure 3, it is still of great value to design a fine-grained pruning method that offers deeper insight into which parameters are critical or redundant, thereby enabling a better trade-off between efficiency and generation quality.

To address these, we propose a tri-level pruning scheme combined with knowledge alignment, enabling us to derive an efficient DiT architecture from a larger teacher model. Our approach maintains competitive performance while meeting the requirements for edge deployment.

### 4.2.1 TRI-LEVEL PRUNING

As exhibited in Figure 3, the transformer block pruning is a simple yet coarse approach, we consider it a low-granularity pruning. To enable finer granularity and better address redundancies, we propose a tri-level pruning scheme that incorporates block pruning and further introduces fine-grained pruning techniques, including head pruning for the multi-head attention mechanism and channel pruning for the linear layer. Notably, the pruned model can be converted into a dense and compact form, enabling execution on mobile devices without requiring additional compilation or specialized hardware support.

We employ a set of learnable binary masks to implement the tri-level pruning scheme. Each binary mask encodes the importance of its corresponding granularity, *i.e.* *block*, *attention-head*, or *linear dimension*. A mask value of 0 indicates that the corresponding unit should be pruned, while a value of 1 denotes that it should be preserved. Specifically, block pruning can be formulated as shown in Eq. (3), where $y_{b_i}, x_{b_i}$ indicate the input and output features of the $b_i^{\text{th}}$ DiT block, $m_{b_i} \in \{0,1\}$ is the binary mask associated with that block, and $\mathcal{M}_b = [m_{b_1}, \ldots, m_{b_N}] \in \{0,1\}^N$ denotes the set of binary masks for all DiT blocks. When $m_{b_i} = 1$, the block is active; otherwise, its output is bypassed through a residual connection.

$$y_{b_i} = \text{Block}_{b_i}(x_{b_i}) \odot m_{b_i} + x_{b_i} \odot (1 - m_{b_i}), \quad m_{b_i} \in \{0,1\}, \mathcal{M}_b = [m_{b_1}, \ldots, m_{b_N}] \in \{0,1\}^N \quad (3)$$

The other two pruning schemes can be expressed using a unified formulation, since pruning *attention heads* is equivalent to removing specific output features before the multi-head attention operation for each token. By integrating the pruning mechanism into the linear layer, the operation can be formulated as shown in Eq. (4):

$$y_{l_i} = \text{Linear}_i(x_{l_i}, W_{l_i}, b_{l_i}) \odot \mathcal{M}_{l_i}, \quad m_{l_i}^d \in \{0,1\}, \mathcal{M}_{l_i} = [m_{l_i}^1, \cdots, m_{l_i}^D] \in \{0,1\}^D \quad (4)$$

where $y_{l_i}, x_{l_i}$ denote the output and input features of the $l_i^{\text{th}}$ linear layer, and $\mathcal{M}_{l_i} \in \{0,1\}^D$ is a binary mask with $D$-dimension corresponding to the output channels. For each $m_{l_i}^d \in M_{l_i}$, a value of $m_{l_i}^d = 0$ zeros out the corresponding output channel at dimension $d$ for layer $l_i^{\text{th}}$; otherwise the channel remains active.

The proposed tri-level pruning scheme begins by generating a candidate mask set $\mathbb{M}$ for each pruning target, as illustrated in Figure 2. These candidate masks are selected based on the desired number of

active components, which are constrained by the memory limitation of the target device. Notably, exhaustively exploring all pruning combinations results in an extremely large search space, making the optimization problem intractable (*e.g.*, pruning 6 out of 32 attention heads results in 906,192 possible configurations). To mitigate this, we adopt a group-wise masking mechanism that partitions overall search space into smaller subspaces, allowing pruning to be performed efficiently within each subspace. Once the candidate masks are generated, we further optimize them to identify the optimal configuration that minimizes the information loss caused by pruning.

### 4.2.2 KNOWLEDGE DISTILLATION VIA FEATURE ALIGNMENT

Knowledge distillation (KD) is a widely adopted technique for transferring knowledge from a teacher model to a student model. Therefore, it is an effective strategy for preserving the performance of the pruned model. However, due to varying pruning schemes, the pruned student model may have different feature widths compared to the teacher model, which poses challenges for traditional distillation. Inspired by Yu et al. (2025), we employ a trainable affine transformation to align the features between the teacher and the student model. Thus, distillation is then performed using the aligned features. This process is formally defined in Eq. (5), where $y_{t_i}$ and $y_{s_i}$ represent the output features of $i^{\text{th}}$ DiT block group for the teacher model and the student model respectively:

$$\mathcal{L}_{distill} = \frac{1}{N} \sum_{i=1}^{N} \text{sim}(y_{t_i}, \mathcal{F}_i(y_{s_i}; \Theta_i));$$

(5)

Here, $N$ denotes the number of DiT block groups, and $\text{sim}(\cdot, \cdot)$ is a similarity alignment function used to match the feature distributions between teacher and student. The function $\mathcal{F}_i(\cdot; \Theta_i)$ is an affine transformation parameterized by $\Theta_i$, introduced to align the dimensionality of the student features with that of the teacher. The overall training loss is formulated as Eq. (6), where $\mathcal{L}_{\text{flow-matching}}$ is the conditional flow-matching objective from Eq. (2), and $\alpha$ is a hyper-parameter to adjust the weight of distillation. $\alpha$ is set to 0.01 in our experiments.

$$\mathcal{L} = \mathcal{L}_{\text{fm}} + \alpha \mathcal{L}_{\text{distill}}$$

(6)

### 4.2.3 INTEGRATION WITH HARDWARE-AWARE OBJECTIVE

Here, we specify the details of our tri-level pruning scheme for constructing an efficient Diffusion Transformer architecture tailored to the iPhone 16 Pro Max. Due to the memory limitation of the device, the total number of parameters must remain under 1 billion. Based on the sensitivity analysis in Figure 3, which indicates that FFN contributes more significantly to performance than attention heads, we prioritize pruning attention heads more aggressively. Starting from a 2B parameter base model with 28 DiT blocks, 32 attention-heads, and FFN dimension of 8192, our final efficient model archives 915M parameters by pruning 8 blocks, 12 attention heads, and reducing the FFN dimension by 25% following Algorithm 1. More details can be found in Appendix B.

### 4.3 ADVERSARIAL FINE-TUNING FOR STEP DISTILLATION

We adopt adversarial fine-tuning of step-distillation, following Wu et al. (2025b), with a generator generator $\mathcal{G}_\theta(t, \mathbf{x}_t)$ and a discriminator $\mathcal{D}\phi(t, \mathbf{x}_t)$. The generator $\mathcal{G}_\theta$ is initialized with the pretrained DiT denoiser weights. The discriminator $\mathcal{D}\phi$ inherits the DiT backbone, whose first $K$ blocks are initialized from $\mathcal{G}_\theta$ and frozen to serve as a timestep-conditioned feature parser, while the subsequent DiT block is learnable and include 3D self-attention and cross-attention to enhance spatio-temporal capacity. An MLP head with SiLU activation is appended to produce real/fake logits as shown in Figure 2. The generator $\mathcal{G}_\theta$ learns to generate clean samples in a few steps (*i.e.* 4-step), while the discriminator $\mathcal{D}\phi$ distinguishes real/generated samples, see Appendix C for details. The adversarial fine-tuning reduces diffusion sampling budget by up to $20\times$ comparing to full-step diffusion.

### 4.4 TILED GEMM FOR EFFICIENT FFN INFERENCE

In Transformers, the Feed-Forward Network (FFN) is a token-wise two layer MLP applied after the attention mechanism. It expands the channel dimension from $d$ to $Nd$ (typically $N \in [2, 4]$) and projects back to $d$ via a nonlinearity activation layer (*e.g.* SiLU). The design increases the expressive capacity of the token-wise mapping, raising the effective rank and refining token-level representation, while attention primarily mixes information across tokens.

While the expansion in FFN ($d \rightarrow Nd \rightarrow d$) improves quality, the large General Matrix to Matrix Multiplications (GEMMs) become a memory bottleneck on mobile devices. Although our KD-guided pruning (Sec. 4.2) and adversarial step distillation (Sec. 4.3) reduce overall complexity, GEMMs in FFN remain limited by the device's memory bandwidth. Addressing this bottleneck with conventional kernel-level optimization is infeasible as the deployment compiler, Apple's CoreML, is a closed-source tool. Therefore, we introduce an operator-level tiled GEMM strategy for the $d \rightarrow Nd \rightarrow d$ projections as illustrated in Figure 5. This method partitions the weight matrix along the expansion dimension $Nd$ into smaller, cache-friendly tiles, and activation are fused within each tile to reduce extra reads/writes. This design mitigates I/O bottleneck, improving cache usage and alleviating bandwidth pressure, particularly for large feature dimensions such as $d=8192$.

We benchmark the latency of a fixed number of input tokens ($L=2048$) and $N=4$ while varying $d$, comparing a naïve implementation to the tiled GEMM as shown in Figure 6. For reference we also plot the theoretical scaling estimated from FLOPs, $4LNd^2$. As $d$ increases, the naïve GEMM's latency grows faster than the theoretical baseline, indicating a memory-bound issue, whereas the tiled GEMM remains close to the theoretical baseline, demonstrating reduced memory traffic.

The tiled-GEMM in FFN yields an $\sim$10% end-to-end DiT forward speedup on-device, without fine-tuning and complementary to the KD-guided pruning and step-distillation benefit.

## 5 EXPERIMENTS

**Training.** We train on both curated real-world image/video data and synthetic data. We use $128$ NVIDIA A100 80GB GPUs for DiT training, using AdamW optimizer with $5e-5$ learning rate and betas values as $[0.9, 0.999]$. We build our Diffusion Transformer following public models (Yang et al., 2024b; HaCohen et al., 2024b), and incorporate QK normalizations and Rotary Positional Embeddings (RoPE) (Su et al., 2024). The T5 text encoder (Raffel et al., 2020) is employed to capture textual information. The training is conducted using low resolution image and video for pretraining and then finetuning with high resolution data. More training details in Appendix D.

**Adversarial Fine-tuning** is conducted for $20K$ iterations on $64$ A100 GPUs, using the AdamW optimizer with a learning rate of $1e-6$ for the generator (*i.e.*, DiT) and $1e-4$ for the discriminator heads. We set the betas as $(0.9, 0.999)$ for both the generator and the discriminator optimizers. We set the EMA rate as $0.95$ following Zhang et al. (2024). Additional details are reported in Appendix D.

**Evaluation and Deployment.** Our models are evaluated following the standard Vbench (Huang et al., 2024) setting, that is, we generate 5 videos for each prompt, and test the scores over the 1K prompt set. Both server and mobile-deployed models are step-distilled and evaluated with 4-step generation. The server model generates 121-frame horizontal videos at a resolution of $576 \times 1024$, without classifier-free guidance. The generated video is saved at 5 seconds 24 FPS for score testing and qualitative visualization. We use different seeds and find the $\Delta$VBench score is lower than $\pm 0.2$. For mobile demo, we generate $49 \times 384 \times 512$ videos on iPhone 16 Pro Max using CoremlTools (Apple Inc., 2024) under half precision. We employ the CLIP text encoder for on-device text encoding efficiency, while the T5 encoder is utilized for the server-side model. Additional details are in Appendix J. We measure the latency by 50 runs and take the median.

### 5.1 QUALITATIVE RESULTS

We visualize our generated videos in Figure 4. Our model consistently produces high-quality video frames and smooth object movements. To demonstrate the generic text-to-video generation ability, we show various generation examples, including human, animal, photorealistic, and art-styled scenes. We include more video visualizations and the efficient model demo in *supplementary material*.

### 5.2 QUANTITATIVE BENCHMARK

We evaluate our method on VBench (Huang et al., 2024) and compare it against recent state-of-the-art DiT-based video generation models, as shown in Tab. 2. Although our model is compact and optimized for fast inference on mobile devices, it achieves a higher total score than several strong baselines, including the OpenSora-1.2, CogVideoX-2B (Yang et al., 2024b), LTX-Video (HaCohen et al., 2024b). Compared to current open-source SOTA, Wan2.1-1.3B (Wan et al., 2025), our server variant

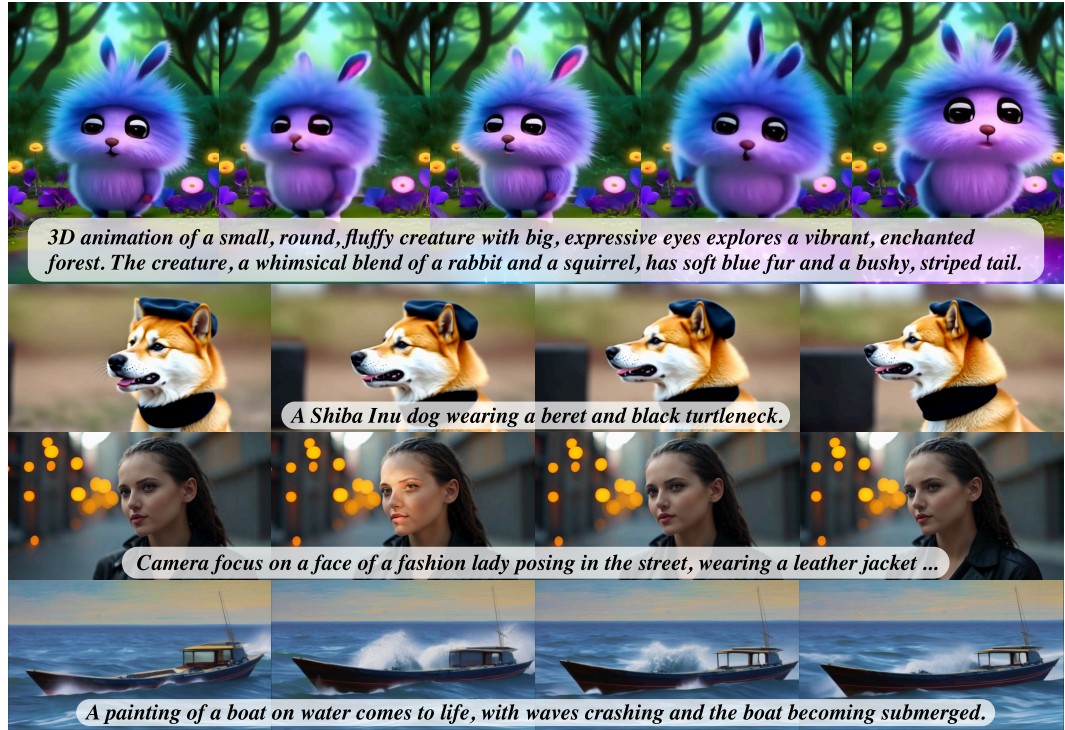

*3D animation of a small, round, fluffy creature with big, expressive eyes explores a vibrant, enchanted forest. The creature, a whimsical blend of a rabbit and a squirrel, has soft blue fur and a bushy, striped tail.*

*A Shiba Inu dog wearing a beret and black turtleneck.*

*Camera focus on a face of a fashion lady posing in the street, wearing a leather jacket ...*

*A painting of a boat on water comes to life, with waves crashing and the boat becoming submerged.*

Figure 4: Video generated by our efficient diffusion transformer.

Table 2: VBench (Huang et al., 2024) comparison with popular open-source Diffusion Transformer video generation models. Scores for open-source models are collected from the VBench Leaderboard.

| Model | Params (B) | Total Score | Quality Score | Semantic Score | Temporal Flickering | Aesthetic Quality | Imaging Quality | Object Class | Scene | Overall Consistency | Dynamic Degree | Motion Smoothness |
|---|---|---|---|---|---|---|---|---|---|---|---|---|
| Wan2.1 | 14 | 84.70 | 85.64 | 80.95 | 99.53 | 61.53 | 67.28 | 94.24 | 53.67 | 27.44 | 94.35 | 96.92 |
| Wan2.1 | 1.3 | 83.31 | 85.23 | 75.65 | 99.55 | 65.46 | 67.01 | 88.81 | 41.96 | 25.50 | 65.19 | 98.52 |
| Open-Sora-2.0 | 11 | 84.34 | 85.40 | 80.12 | 99.40 | 64.39 | 65.66 | 94.50 | 52.71 | 27.50 | 71.39 | 98.69 |
| Open-Sora-1.2 | 1.2 | 79.76 | 81.35 | 73.39 | 99.53 | 56.85 | 63.34 | 82.22 | 42.44 | 26.85 | 42.39 | 98.50 |
| Hunyuan | 13 | 83.24 | 85.09 | 75.82 | 99.44 | 60.36 | 67.56 | 86.10 | 53.88 | 26.44 | 70.83 | 98.99 |
| CogVideoX1.5 | 5 | 82.01 | 82.72 | 79.17 | 98.53 | 62.07 | 65.34 | 83.42 | 53.28 | 27.42 | 56.16 | 98.15 |
| CogVideoX | 5 | 81.91 | 83.05 | 77.33 | 78.97 | 61.88 | 63.33 | 85.07 | 51.96 | 27.65 | 69.51 | 97.20 |
| CogVideoX | 2 | 81.55 | 82.48 | 77.81 | 98.85 | 61.07 | 62.37 | 86.48 | 50.04 | 27.33 | 97.51 | 64.79 |
| Step-Video | 30 | 81.83 | 84.46 | 71.28 | 99.40 | 61.23 | 70.63 | 80.56 | 24.38 | 27.12 | 53.06 | 99.08 |
| Mochi-1 | 10 | 80.13 | 82.64 | 70.08 | 99.40 | 56.94 | 60.64 | 86.51 | 36.99 | 25.15 | 61.85 | 99.02 |
| LTX-Video | 1.8 | 80.00 | 82.30 | 70.79 | 99.34 | 59.81 | 60.28 | 83.45 | 51.07 | 25.19 | 54.35 | 98.96 |
| Ours-Server | 2.0 | 83.09 | 84.65 | 76.86 | 98.74 | 64.72 | 65.85 | 90.57 | 52.76 | 27.28 | 65.28 | 99.21 |
| Ours-Mobile | 0.9 | 81.45 | 83.12 | 74.76 | 98.11 | 64.16 | 63.41 | 92.26 | 51.06 | 25.51 | 58.33 | 99.23 |

achieves comparable quality while delivering faster inference speed per sampling step. Importantly, the mobile deployment (0.9B parameters) maintains competitive scores relative to larger models while running efficiently on the iPhone 16 Pro Max. These results highlight the effectiveness of our DiT pruning and distillation method. Human evaluation studies further demonstrate the perceptual quality of our models as reported in Appendix A.

**Comparison with Other Mobile Video Generation Methods.** We compare the VBench score of our model against SnapGen-V (Wu et al., 2025b) and on-device Sora (Kim et al., 2025), using benchmark metrics reported in their paper to show the performance of our mobile-deployed model as in Tab. 3. Notably, SnapGen-V is based on UNet architecture for efficient video generation, and on-device Sora is a training-free method that enables open-sora (Zheng et al., 2024) on the mobile device, thus we include its baseline, OpenSora v1.2 as reference. Our mobile model outperforms SnapGenV by +0.31 total score and outperforms on-device Sora by a large margin in dynamic degree,

demonstrates significantly improved motion. These results demonstrate that our approach provides competitive or superior performance compared to existing mobile video generation methods.

Table 3: Comparison with current on-device diffusion video generation models.

| Model | Total Score | Quality Score | Semantic Score | Subject Consistency | Background Consistency | Temporal Flickering | Motion Smoothness | Dynamic Degree | Aesthetic Quality | Imaging Quality |
|---|---|---|---|---|---|---|---|---|---|---|
| Ours-Mobile | 81.45 | 83.12 | 74.76 | 95.73 | 96.64 | 98.11 | 99.23 | 58.33 | 64.16 | 63.41 |
| SnapGen-V | 81.14 | 83.47 | 71.84 | – | – | 99.37 | – | 51.11 | 62.19 | – |
| Open-Sora V1.2[*] | 79.76 | 81.35 | 73.39 | 96.75 | 97.61 | 99.53 | 98.50 | 42.39 | 56.85 | 63.34 |
| On-device Sora[†] | – | – | – | 96.00 | 97.00 | 99.00 | 99.00 | 27.00 | 47.00 | 53.00 |

[*]We report the result of open-sora for reference since on-device Sora reported result precision and scale differ from the standard VBench values.
[†]We converted reported on-device Sora's VBench values to common scale for comparison.

Table 4: Scaling VAE compression ratio. VAE PSNR is measured on DAVIS (Perazzi et al., 2016) with $33 \times 512 \times 512$ resolution. Latencies are for one denoising step. VBench scores are provided.

| VAE | | Diffusion Transformer | | | | | | |
|---|---|---|---|---|---|---|---|---|
| Compression Factor | PSNR | Latency (ms) | Total | Quality | Semantic | Aesthetic | Consistency | Flickering |
| $4 \times 16 \times 16$ | 33.1 | 7900 | 80.35 | 82.05 | 73.54 | 64.45 | 26.80 | 98.59 |
| $4 \times 32 \times 32$ | 30.9 | 920 | 79.95 | 82.99 | 67.83 | 61.52 | 27.07 | 97.46 |
| $8 \times 32 \times 32$ | 30.6 | 380 | 79.80 | 82.59 | 68.66 | 61.80 | 27.17 | 97.70 |
| $8 \times 64 \times 64$ | 28.2 | 90 | 78.40 | 81.79 | 64.86 | 55.29 | 26.11 | 97.52 |

**Relation of VAE Compression and DiT Performance.** According to the benchmark on Tab. 4 , higher VAE compression ratio degrade the reconstruction of high-frequency details (*e.g.,* texture), which directly impacts the aesthetic score. Furthermore, the reconstruction losses introduced by aggressive compression can make generated objects harder for the encoder, utilized in VBench, to recognize, leading to lower semantic scores. In contrast, temporal flickering and temporal consistency remain largely robust. This robustness can be attributed to the use of 3D convolutions in VAE, which effectively preserve temporal coherence, thereby maintaining high scores even under higher compression. We show that $8 \times 32 \times 32$ compression ratio hits a sweet point in quality-speed tradeoff for mobile use case.

## 5.3 ABLATION STUDY

**Scaling VAE Compression Ratio.** In Tab. 4, we scale the VAE compression ratio and compare DiT generation speed and video quality. For a fair comparison, we train a 2B-parameter DiT with each VAE and measure per-step generation speed by testing one denoising step on the Nvidia A100 GPU at $121 \times 576 \times 1024$ resolution. We observe that though lower compression VAEs ($4 \times 16 \times 16$) can achieve better reconstruction PSNR, the generation speed is slower by magnitudes. On the other hand, aggressive compression ($8 \times 64 \times 64$) results in poor reconstruction, and will negatively impact generation quality (*i.e.* VBench scores). We find that ($8 \times 32 \times 32$) hits a balance between speed and quality, and employ this configuration for our Diffusion Transformer. The training details and more experiment results for the video VAE can be found in Appendix F and G.

**Impact of KD-Guided Training.** We evaluate the effect of the proposed KD-guided training via ablation with and without distillation. As shown in Tab. 5, the tri-level trained with the proposed distillation objective consistently outperforms the counterpart without distillation, indicating that it helps recover capacity lost due to pruning.

**Impact of Tri-Level Pruning.** We compare the proposed tri-level pruning against random masking and block-only pruning (*shallow*) following Fang et al. (2024); Xie et al.. All variants are initialized from the same pretrained 2B DiT. As shown in Tab. 5, the tri-level pruned model consistently outperforms both random-masked and *shallow* baseline in Quality and Total scores. These results indicate the proposed tri-level pruning can effectively remove redundant components while preserving capacity, thereby achieving pruning with minimum performance degradation.

**Comparison with Width Pruning.** To further demonstrate the effectiveness of the proposed tri-level pruning, we additionally compare it against width only pruning (*i.e.,* Attention-heads or FFN dimen-

sion). However, width pruning alone is structurally impractical for our specific model design. Our goal is to compress a 2B-parameter model to under 1B parameter (*i.e.,* more than 50% reduction). Since the *qkv-projection* and *out-projection* layers in Self-Attention and Cross-Attention contain $3d^2$ and $d^2$ parameters ($d$ is the model dimension), respectively, and each FFN contains $8d^2$ with an multiplier factor of 4, achieving such a aggressive reduction would effectively require removing entire Attention or FFN, which resulting in a fundamentally broken model. Therefore, we adopt a bi-level pruning baseline that only prunes Attention-heads and FFN dimensions, denoted as the *narrow* model in Tab. 5. The results show that, under the same parameter reduction target, our tri-level pruning consistently outperforms width-only pruning in VBench Scores.

**Comparison with Compact Model Trained from Scratch.** To separate the effect of tri-level pruning and KD-guided training from model size, we train a compact DiT the same model configuration from scratch. As reported in Tab. 5, this compact baseline underperforms the KD-guided tri-level pruned model by large margin across Quality, Semantic, and Total scores. These results indicate that training a small DiT from scratch is suboptimal, while starting from a larger teacher and applying tri-level pruning with knowledge distillation is crucial for preserving overall quality under mobile constraints.

Table 5: Ablation study on tri-level pruning schemes and fine-tuning using proposed knowledge distillation.

| Method | KD | Params(M) | Quality | Semantic | Total |
|--------|-----|-----------|---------|----------|-------|
| tri-level | ✓ | 915 | 83.12 | 74.76 | 81.45 |
| tri-level | ✗ | 915 | 82.19 | 66.23 | 79.00 |
| random | ✗ | 915 | 82.01 | 65.01 | 78.68 |
| *shallow* | ✗ | 932 | 81.63 | 67.15 | 78.73 |
| *narrow* | ✗ | 945 | 80.92 | 65.87 | 77.92 |
| compact[1] | ✗ | 915 | 79.23 | 63.94 | 76.17 |

[1] Train from scratch

Table 6: Ablation study on different discriminator head design. The evaluation is conducted with a 4-step generation without classifier-free guidance.

| Head | #Steps | Quality | Semantic | Total |
|------|--------|---------|----------|-------|
| DiT block + MLP | 4 | 83.81 | 72.89 | 81.63 |
| ResBlock-2D + Temporal-Attn | 4 | 83.24 | 67.78 | 80.14 |
| Lightweight ResBlock | 4 | 80.05 | 66.01 | 77.24 |

**Full Guidance Adversarial Distillation.** We validate the proposed discriminator tailored for the DiT denoiser through an ablation study on the prediction head. The experiments are conducted using our pre-trained 2B parameter DiT model with various discriminator head designs. We compare our design with spatial-temporal heads introduced in Wu et al. (2025b) and the lightweight ResBlock head proposed in Wang et al. (2024a). Since the discriminator head in Wang et al. (2024a) was originally designed for the text-to-image model, we extend it to a Conv3D variant in the ablation. We show that the proposed prediction head (a transformer block followed by an MLP classifier) achieves best 4-step generation performance, with notable gains in semantic scores. We attribute this to improved alignment between the text condition and the hidden states.

## 6 CONCLUSION

In this work, we present an efficient video generation framework that significantly accelerates Diffusion Transformers, making efficient synthesis feasible on mobile devices. By combining a high-compression VAE, latency- and sensitivity-aware pruning, and adversarial step distillation, we successfully deploy DiT video generator to iPhone and reduce inference to just four steps while maintaining high visual quality. Our pipeline achieves over 15 FPS generation speed (generate 49-frame within 4 seconds) on an iPhone 16 Pro Max, demonstrating the practical viability of DiT-based video generation on edge devices. We discuss limitations and broader impact in Appendix N.

## ETHICS STATEMENT

We acknowledge that we have read the Code of Ethics and this manuscript complies with it.

## REPRODUCIBILITY STATEMENT

Training settings and datasets are reported in Appendix D and F. The pruned architecture and search constraints needed to reproduce the mobile model are reported in Sec. 4.2 and Appendix B and D. Technical details to reproduce tiled GEMM are reported in Sec. 4.4 and Appendix E. Evaluation follows VBench official guideline to benchmark text-to-video generation quality.

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

## A  USER STUDY

To evaluate the human preference across different video generation models, we conduct human evaluations comparing our model against baselines such as CogVideoX-2B, LTX-Video in Tab. 7. We generate video clips using prompts from VBench and MovieGenBench (Polyak et al., 2024) and ask human labelers to select the best results across prompt alignment, aesthetics, and motion quality. The results indicate that our model significantly outperforms the baselines.

To further demonstrate the effectiveness of our proposed method, we also evaluate the human preferences across our server-side model and mobile-deployed model in Tab. 8. We generate video samples in $49 \times 512 \times 384$ for both models. The result illustrates the trade-off in visual quality in terms of efficiency. Notably, our mobile-deployed model uses CLIP as text-encoder for efficiency while the server-side model uses T5-Encoder.

Table 7: Human preference across different video generation models.

| Model | Prompt Alignment | Aesthetics | Motion Quality |
|---|---|---|---|
| LTX-Video[*] | 16.7% | 10.0% | 6.7% |
| CogVideoX-2B | 40.0% | 33.3% | 43.3% |
| Ours | 43.3% | 56.7% | 50.0% |

[*]We notice the performance of LTX-Video highly depends on the prompts enhancement.

Table 8: Human preference across our sever-side and mobile-deployed model.

| Model | Prompt Alignment | Aesthetics | Motion Quality |
|---|---|---|---|
| Server-side | 58.8% | 52.9% | 55.8% |
| Mobile-deployed | 41.2% | 47.1% | 44.1% |

## B  SEARCH FOR OPTIMAL PRUNING CONFIGURATION

We describe the detail to determine the optimal pruning configuration for on-device model as Algorithm 1. Starting from a pretrained DiT, our objective is to satisfy the device memory budget while minimizing quality degradation.

**Search space and constraints.** We enumerate candidate configurations over three granularities: Attn-Heads $\in \{20, 24, 28, 32\}$, Blocks $\in \{16, 20, 24, 28\}$, FFN dim $\in \{5120, 6144, 7168, 8192\}$. Here, values denote the *kept* attention heads per block, transformer blocks, and FFN dimension, respectively. To reduce the search space, we ignore extreme cases (*e.g.* pruning $> 50\%$ of blocks).

**Sensitivity-guided pruning order.** Given by the sensitivity analysis in Figure 3, we rank pruning sensitivity as: FFN dim > Blocks > Attn-Heads. Accordingly, we prefer to prune more aggressively on attention heads and conservatively on FFN dimension, with blocks in between.

For each candidate $c$, we initialize all possible binary masks and optimize them with the KD-guided tri-level pruning objectives Eqs. (3), (4) and (6). We employ Gumbel-Softmax sampling for differentiable mask selection. Candidates are evaluated on a fixed validation set using the average flow-matching objective Eq. (2) over multiple timesteps (*i.e.* 25-step), subject to device budget constraints.

## C  ADVERSARIAL FINETUNING FOR STEP-DISTILLATION

To obtain a $k-$step distillation procedure, we predefine the intermediate diffusion timesteps as $\mathcal{T} = \{T_1, T_2, \ldots, T_k\}$ with the following ordering $T_1 = 1 > T_2 > \cdots > T_k > 0$. Typically, $k$ is set to 4 to achieve a $4-$step diffusion model. Given a real data sample $\mathbf{X}_0$, we can obtain the latent $\mathbf{x}_0$ using the VAE. We sample two timesteps $t$ and $t'$ uniformly at random from the set $\mathcal{T}$ such that $t' < t$. We can construct the fake and real samples using the diffusion forward Eq. (1) and velocity from the generator $\mathcal{G}_\theta(t, \mathbf{x}_t)$ as follows:

$$\text{Fake}: \hat{\mathbf{x}}_{t'} = \mathbf{x}_t + (t' - t) \cdot \mathcal{G}_\theta(t, \mathbf{x}_t); \quad \text{Real}: \mathbf{x}_{t'} = (1 - t')\mathbf{x}_0 + t'\epsilon; \epsilon \sim \mathcal{N}(0, I) \qquad (7)$$

Using the above real and fake samples, we can define the discriminator and generator losses. Below, we employ the widely used (Sauer et al., 2023b;a; 2021; Zhang et al., 2024) hinge loss (Lim & Ye, 2017) as the adversarial training objective. The discriminator's goal is to differentiate between real and fake samples by minimizing:

$$\mathcal{L}_{\text{adv}}^{\mathcal{D}} = \mathbb{E}_{t', \mathbf{x}_0} \left[ \text{ReLU}(1 + \mathcal{D}_\phi(\mathbf{x}_{t'}, t')) \right] + \mathbb{E}_{t, t', \mathbf{x}_0} \left[ \text{ReLU}(1 - \mathcal{D}_\phi(\hat{\mathbf{x}}_{t'}, t')) \right], \qquad (8)$$

---

**Algorithm 1** Search for Optimal Pruning Configuration

---

**Require:**
1: $\hat{\epsilon}_\theta^{\text{super}}$: Pretrained DiT.
2: $\mathbb{T}(\cdot)$: A lookup table mapping a configuration $c$ to its parameter count #Params.
3: $\mathcal{A}$: The set of possible pruning actions (*i.e.* prune 4 block, 4 attention heads, and 12.5% FFN dimensions).
4: $D_{\text{val}}$: A fixed validation dataset for evaluating $\mathcal{L}_{\text{MSE}}$.
5: $P_{\text{target}}$: The target parameter count for the final subnet (*i.e.* 1B parameters).
**Ensure:** The optimal subnet configuration: $C_{\text{optimal}}$

6: $\rightarrow$ **Joint search for an optimal subnet:**
7: Initialize current configuration $C_{\text{current}} \leftarrow C_{\text{super}}$ (the largest architecture)
8: **while** $\mathbb{T}(C_{\text{current}}) > P_{\text{target}}$ **do**
9:     *// Evaluate the cost/benefit of all possible pruning actions*
10:     **for** each action $A_i$ in $\mathcal{A}$ **do**
11:         $\Delta\mathcal{L}_{\text{MSE}_i} \leftarrow \text{eval}(\hat{\epsilon}_\theta^{\text{super}}, C_{\text{current}}, A_i)$
12:         $\Delta\#\text{Params}_i \leftarrow \text{GetParamReduction}(C_{\text{current}}, A_i)$
13:     **end for**
14:     *// Execute the most efficient action (least performance drop per parameter removed)*
15:     $\hat{A} \leftarrow \underset{A_i \in \mathcal{A}}{\arg\min} \frac{\Delta\mathcal{L}_{\text{MSE}_i}}{\Delta\#\text{Params}_i}$
16:     Update current configuration: $C_{\text{current}} \leftarrow \text{ApplyAction}(C_{\text{current}}, \hat{A})$
17: **end while**
18: $C_{\text{optimal}} \leftarrow C_{\text{current}}$

19: $\rightarrow$ **Final fine-tuning of the searched architecture:**
20: $\hat{\epsilon}_\theta^{\text{optimal}} \leftarrow \text{GetSubnet}(\hat{\epsilon}_\theta^{\text{super}}, C_{\text{optimal}})$         $\triangleright$ Inherit weights from the super-net
21: Fine-tune $\hat{\epsilon}_\theta^{\text{optimal}}$ with Knowledge Distillation.         $\triangleright$ As described in Sec. 4.2.2

---

The adversarial objective for the generator $\mathcal{L}_{\text{adv}}^{\mathcal{G}}$ and the reconstruction objective $\mathcal{L}_{\text{recon}}$ are defined as:

$$\mathcal{L}_{\text{adv}}^{\mathcal{G}} = \mathbb{E}_{t,t',x_0}[\mathcal{D}_\phi(\hat{\mathbf{x}}_{t'}, t')]; \quad \mathcal{L}_{\text{recon}} = \sqrt{\|\hat{\mathbf{x}}_0 - \mathbf{x}_0\|_2^2 + c^2} - c. \tag{9}$$

where $\hat{\mathbf{x}}_0 = \mathbf{x}_t - t \cdot \mathcal{G}_\theta(t, \mathbf{x}_t)$, and $c > 0$ is an adjustable constant. Following Zhang et al. (2024); Hu et al. (2024), we also incorporate a reconstruction objective to enhance training stability.

# D   Training Details for DiT

**Base Model Configuration.** The base model adopts a standard DiT architecture, incorporating both self-attention and cross-attention mechanisms. It is composed of 28 transformer blocks with a hidden dimension of $d_{\text{model}} = 2048$, and its multi-head attention modules uses 32 heads. The feed-forward network (FFN) employs a multiplier of 4 and uses SiLU activation. The model applies QK-Norm and LayerNorm for normalization, and it also integrates Rotary positional encoding.

**Pre-training.** The DiT training is trained on the internally collected dataset containing high-quality images and videos, which are similar to public large-scale datasets such as Chen et al. (2024). Training is performed in two stages: (i) pretraining on low-resolution (288p) images and videos for $150K$ iterations, followed by (ii) fine-tuning for an additional $50K$ iterations on a mixed-resolution setting (288p and 576p). The KD-guided tri-level pruning is only conducted to obtain the mobile variant after the pretraining stage. The fine-tuning stage for the mobile variant takes $50K$ iterations. Video clips the mobile variant uses is 49-frame clips. All video clips are resampled to 24 fps and cropped to 5-second segments. In the first stage, we only adopt T5 encoder as the text-encoder, leveraging its stronger capacity for modeling long caption and capturing richer text information. During fine-tuning, we additionally incorporate the CLIP text-encoder alongside with T5, since CLIP is the text encoder deployed on-device. Each encoder output is first projected into the DiT latent space, then the projected embeddings are concatenated and fed into the DiT as conditioning. To improve robustness, we randomly mask either the T5 embeddings or the CLIP embeddings during training, enabling better model generalization capacity under both text-encoders.

**KD-guided Tri-level Pruning.** The tri-level pruning procedure is initialized from the fine-tuned DiT model. To determine the pruning ratios at each granularity (blocks, attention heads, and FFN dimensions), we consider both the hardware constrain of the iPhone 16 Pro Max (parameter budget $< 1B$) and the sensitivity analysis in Figure 3 and conduct Algorithm 1. The optimization is run for $20K$ iterations per candidate configuration to obtain a stable pruning scheme that minimizes performance degradation. At the end, We also perform knowledge distillation alone without pruning, where the student is distilled directly from the baseline model, to enhance the performance, with $\alpha = 0.01$ in Eq. (5).

**Adversarial Fine-tuning.** The adversarial fine-tuning is applied to both our server and mobile variants. The generator and discriminator are initialized from the pre-trained model weights. We use a frozen server variant model to generate with classifier-free guidance scale 5 to produce reconstruction targets. During the fine-tuning, the hinge loss is utilized as the adversarial objective, while the reconstruction loss is defined by the $l_2$-Norm between the v-prediction of the generator and the frozen model.

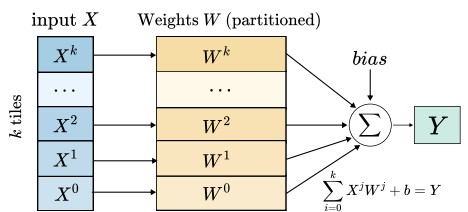
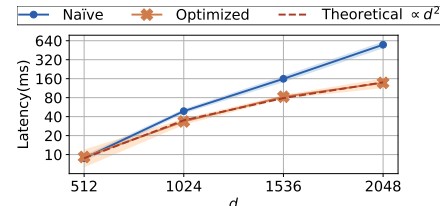

Figure 5: Illustration for tiled GEMM for a single token. The input $X$ and weights $W$ are both tiled into $k$ partitions along input feature.

Figure 6: Latency benchmark for tiled GEMM in FFN. Note that the y axis is $log2$ scale.

## E  IMPLEMENTATION DETAILS IN TILED GEMM

As introduced in Sec. 4.4, Linear Layer with large input feature dimensions tend to become memory-bound. In our implementation, we apply tiled GEMM specifically to the $Nd \rightarrow d$ projection layer in the FFN, which we find to be the dominant bandwidth bottleneck. Other linear layers, such as QKV projections in attention, have much smaller hidden dimensions and show only marginal improvements with tiling, thus we retain their standard implementation.

In pratical, we set the number of partition to $k = 4$, which we identify as a practical balance between parallelism and cache locality on the iPhone 16 Pro Max. This operator-level strategy allows us to reduce data traffic without requiring changes to CoreML's kernel backend, making it directly hardware friendly on mobile device.

## F  TRAINING DETAILS FOR VAE

The VAE is trained on the internal dataset using 64 NVIDIA A100 80GB GPUs. The model is optimized using the AdamW optimizer with $1e - 4$ learning rate, $\beta = (0.9, 0.999)$, and trained with a batch size of 16 per GPU. The training process including two-stage: we first pretrain the model on $256 \times 256$ cropped 17-frame clips for $100K$ iterations, then fine-tuning on a range of resolutions and clip lengths for additional $50K$ iterations.

The VAE is optimized using a combination of a reconstruction loss between the input data and the reconstruction output, and KL-divergence regularization that enforces the latent distribution to follow a normal distribution, as defined in Eq. (10). The KL weight is set to $\lambda = 1e - 6$.

$$\mathcal{L}_{\text{recon}} = \|\hat{y} - y\|$$
$$\mathcal{L}_{\text{KL}} = \tfrac{1}{2}\left[-\log(\sigma^2) - 1 + \sigma^2 + \mu^2\right] \tag{10}$$
$$\mathcal{L} = \mathcal{L}_{\text{recon}} + \lambda\mathcal{L}_{\text{KL}}$$

## G  MORE EXPERIMENTS ON VAE VARIANTS

We further analyze the impact of the number of latent channel to the VAE and the corresponding DiT performance, as shown in Tab. 9. As expected, increasing latent channels is always beneficial for VAE reconstruction quality. However, the benefit quickly saturates, *i.e.* for higher compression VAEs, the reconstruction PSNR can not match that of low-compression VAEs simply by increasing latent channels. Moreover, larger latent channels can harm or destabilize diffusion quality, as display in the $|z| = 512$ setting. Therefore, we select latent channel for each compression setting according to the diffusion quality (*i.e.* VBench score).

Beyond the number of latent channels, we also investigate the impact of patchify options on the overall data dimension compression strategy. Here, we compare direct compression of video data against those use lower-compression VAEs combined with patchify operations later, as shown in Tab. 10. The latter approach achieves higher reconstruction PSNR due to compression is applied more conservatively along the spatial and temporal dimensions. For a fair comparison, we keep the number of latent channels same for overall compression strategy after patchify.

The reconstruction PSNRs are evaluated on DAVIS with $33 \times 512 \times 512$ resolution, while VBench score are evaluated following the standard setup.

Table 9: Impact of VAE latent channels $|z|$.

| VAE | $|z|$ | PSNR | VBench |
|---|---|---|---|
| $4 \times 16 \times 16$ | 64 | 33.1 | 80.35 |
| $4 \times 16 \times 16$ | 128 | 33.6 | 80.33 |
| $8 \times 32 \times 32$ | 128 | 30.6 | 79.73 |
| $8 \times 32 \times 32$ | 256 | 30.8 | 79.80 |
| $8 \times 64 \times 64$ | 128 | 26.9 | 74.43 |
| $8 \times 64 \times 64$ | 256 | 28.2 | 78.40 |
| $8 \times 64 \times 64$ | 512 | 28.5 | 78.00 |

Table 10: Impact of VAE latent channel and patchify size for different compression strategy.

| VAE | Patchify size | $f$ | $|z|$ | PSNR | VBench |
|---|---|---|---|---|---|
| $4 \times 8 \times 8$ | 2 | 1024 | 16 | 33.2 | 80.32 |
| $4 \times 16 \times 16$ | 1 | | 64 | 33.1 | 80.35 |
| $4 \times 8 \times 8$ | 4 | | 16 | 33.2 | 80.08 |
| $4 \times 16 \times 16$ | 2 | 4096 | 64 | 33.1 | 80.19 |
| $4 \times 32 \times 32$ | 1 | | 256 | 30.9 | 79.95 |

## H  IMPACT OF INFERENCE STEPS

Our distilled model supports generation with reduced number of inference steps. We further exhibits the impact of the inference steps for the generation results in Tab. 11.

Table 11: Ablation study on different inference steps.

| #Steps | Quality | Semantic | Total |
|---|---|---|---|
| 1 | 74.95 | 64.41 | 72.84 |
| 2 | 78.70 | 69.92 | 76.94 |
| 4 | 83.81 | 72.89 | 81.63 |
| 8 | 83.89 | 73.12 | 81.74 |

## I  FINETUNE LTX-VIDEO ON INTERNAL DATASETS

To mitigate the potential impact of training with internal datasets Sec. 6, we trained the LTX-Video model using our internal dataset, and the VBench score is shown in Tab. 12. The results demonstrate that our internal dataset achieves on-par performance comparing to the original LTX-Video dataset.

Table 12: The VBench score of official LTX-Video and trained using our internal datasets.

| Model | Total | Quality | Semantic | Aesthetics | Scene | Consistency |
|---|---|---|---|---|---|---|
| LTX-Video | 80.00 | 82.30 | 70.79 | 59.81 | 83.45 | 25.19 |
| LTX-Video (Ours) | 80.35 | 82.05 | 73.54 | 64.45 | 37.08 | 26.80 |

## J  MOBILE DEPLOYMENT

We deploy our model on an iPhone 16 Pro Max by converting to FP16 and executing on the Neural Engine and the CPU cores. To improve on-device numerical stability, we adopt HardSiLU as the activation function and LayerNorm for normalization. For text encoding, we employ the CLIP text encoder for on-device efficiency, while the T5 encoder is still utilized for the server-side model.

## K  LATENCY BENCHMARK RESULTS ON MOBILE

We provide screenshots illustrating the latency of our mobile model. The latency is benchmarked on an iPhone 16 Pro Max using Apple's CoreML toolkits within Xcode. The reported latency corresponds to the median value collected across multiple runs. The model is chunked to two chunks for efficient loading and inference. As shown in Figure 7, the inference time for one-step DiT model is $668.02$ ms in total. Accordingly, our 4-step model requires $3,021$ ms to generate a 49-frame video at $512 \times 384$ resolution.

The latency breakdown for each components in the generation pipeline is shown in the Tab. 13. Latency for module loading and diffusion backward process are included in I/O and Misc. The overall latency for the whole pipeline is $3,318$ ms, resulting in an average generation speed of $15$ FPS.

Table 13: Latency breakdown for on-device demo.

| Module | Text Encoder | DiT | VAE decoding | I/O and Misc |
|---|---|---|---|---|
| Latency (ms) | 6 | 668[†] | 230 | 320 |

[†]The latency for the DiT is corresponds to a single denoising step.

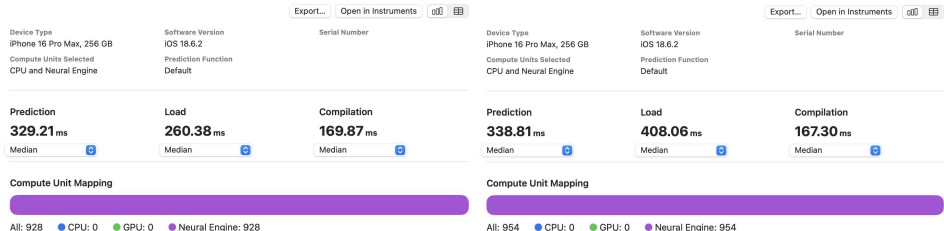

Figure 7: Latency Benchmark on Apple's CoreML toolkits within Xcode.

## L  FULL VBENCH SCORES OF OUR MODELS

For completeness, we additionally report the full set of VBench scores for our models to enable a thorough comparison as shown in Tab. 14.

## M  USE OF LLMS

We used large language models (*e.g.* ChatGPT, Gemini) solely to assist with manuscript formatting. No part of the research design, experimental implementation, or analysis relied on LLMs.

## N  LIMITATIONS AND BROADER IMPACT

Despite these advances, our method has several limitations. First, the highly compressed latent space and DiT pruning lead to occasional degradations in fine-grained details, particularly in fast motion or complex texture scenes. Second, due to various practical constraints, most state-of-the-art video diffusion models (VDMs) used for comparison in this work, including our own, are trained on internally collected video datasets that cannot be fully disclosed or released. As a result, direct comparisons may not be entirely fair and reproducible. To mitigate this limitation, we include a

Table 14: Full VBench Scores of our model for further comparison

| Model | Total Score | Quality Score | Semantic Score | Subject Consistency | Background Consistency | Temporal Flickering | Motion Smoothness | Dynamic Degree | Aesthetic Quality | Imaging Quality | Object Class | Multiple Objects | Human Action | Color | Spatial Relationship | Scene | Appearance Style | Temporal Style | Overall Consistency |
|---|---|---|---|---|---|---|---|---|---|---|---|---|---|---|---|---|---|---|---|
| Ours-Server | 83.09 | 84.65 | 76.86 | 96.48 | 97.32 | 98.74 | 99.21 | 65.28 | 64.72 | 65.85 | 90.57 | 58.38 | 96.60 | 87.35 | 69.47 | 52.76 | 22.87 | 25.54 | 27.28 |
| Ours-Mobile | 81.45 | 83.12 | 74.76 | 95.73 | 96.65 | 98.11 | 99.23 | 58.33 | 64.16 | 63.41 | 92.26 | 55.02 | 95.00 | 84.81 | 66.85 | 51.06 | 22.68 | 24.51 | 25.51 |

reproduction of the LTX model trained on our dataset and report the results in the Appendix I. Third, we evaluate on-device performance exclusively on iPhone devices. We focus on this platform because it offers a more streamlined implementation pipeline compared to Android mobile devices, and it follows the common practice adopted by prior works such as Zhao et al. (2024); Li et al. (2023); Wu et al. (2025b). Extended evaluation on a broader range of mobile platforms is an important direction for future work.

## O  MORE QUALITATIVE RESULTS

We illustrate more qualitative results of video clips generated by our model in Figures 8 and 9.

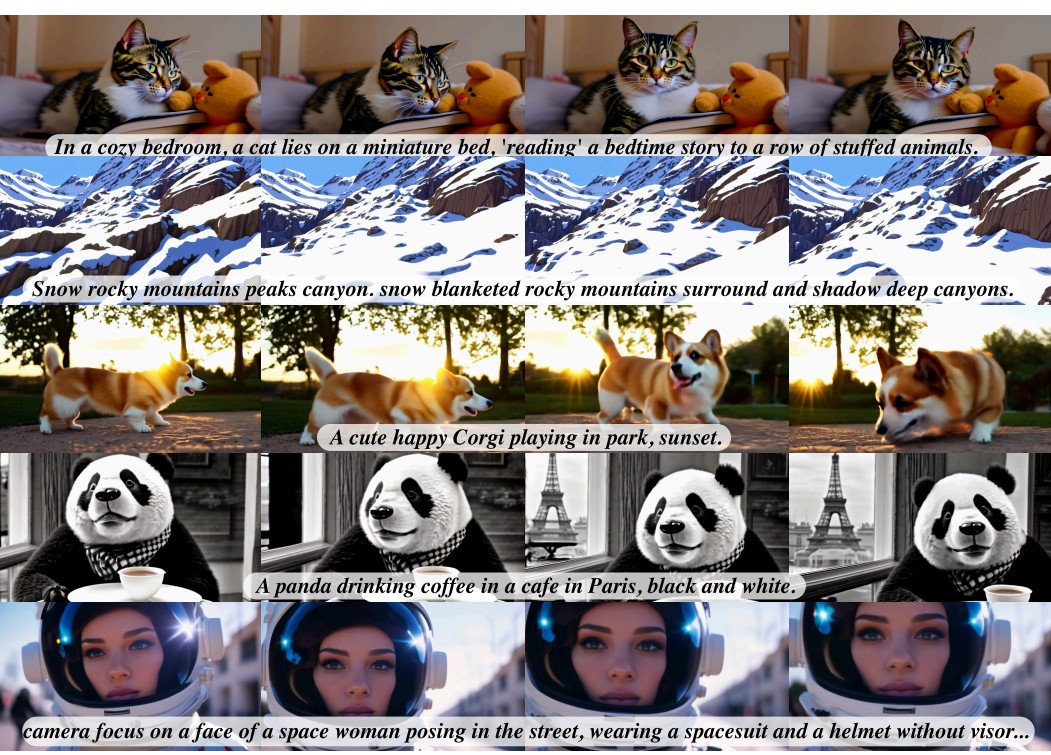

*In a cozy bedroom, a cat lies on a miniature bed, 'reading' a bedtime story to a row of stuffed animals.*

*Snow rocky mountains peaks canyon. snow blanketed rocky mountains surround and shadow deep canyons.*

*A cute happy Corgi playing in park, sunset.*

*A panda drinking coffee in a cafe in Paris, black and white.*

*camera focus on a face of a space woman posing in the street, wearing a spacesuit and a helmet without visor...*

Figure 8: More quality results generated by our server model.

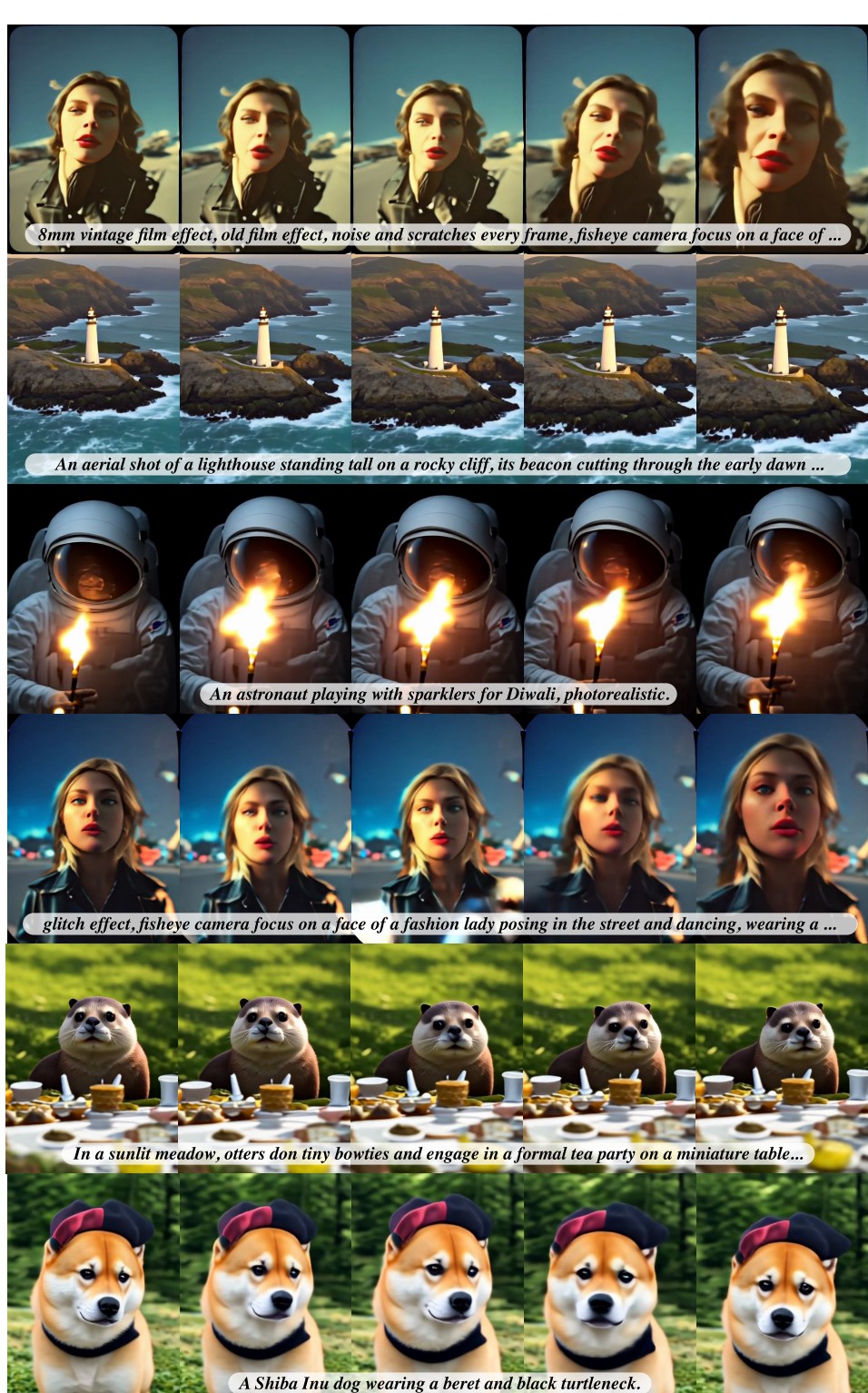

Figure 9: More quality results generated by our mobile model.

