# OpenReview forum: "Taming Diffusion Transformer for Efficient Mobile Video Generation in Seconds"
_ICLR.cc/2026/Conference — Submitted to ICLR 2026_

### Official Review · Reviewer_XXHh · 2025-10-27

**Soundness:** 3
**Presentation:** 3
**Contribution:** 2
**Rating:** 4
**Confidence:** 4

**Summary:**

This paper presents several modifications to DiT-based video generation models to enable them to run on mobile devices. Specifically, for the model architecture, the paper adopts a high-compression video VAE to reduce the number of tokens and a lightweight DiT obtained through structured pruning. For training, adversarial distillation is used to reduce the number of timesteps required during inference. For efficient inference, a tiled GEMM strategy is applied to alleviate memory bottlenecks. The experimental results show that the proposed model can generate a 49-frame video clip within 4 seconds on an iPhone 16 Pro Max.

**Strengths:**

- The motivation for this work is clear and convincing. Nowadays, many video generation models or services have been released, but almost all of them require a huge amount of computational resources to run. Reducing this cost could have a significant impact on user-generated content creation and also contribute to greater sustainability.
- The base model used in this work, DiT in latent space, is a popular choice for video generation tasks. Therefore, the proposed modifications are likely to be widely applicable to existing or future video generation models.

**Weaknesses:**

- The novelty in methodology is marginal.
  - Increasing the compression ratio in VAE is a common choice (e.g., [R1]) for efficient latent diffusion models. To enhance the novelty, it would be beneficial if the authors could provide empirical findings that are particularly important in the context of video generation.
    - [R1] "SANA: Efficient High-Resolution Image Synthesis with Linear Diffusion Transformers," ICLR 2025.
  - The idea of tri-level pruning sounds interesting, but the optimization process is greedy and not specifically designed for tri-level pruning. If the authors want to argue for the benefits of the tri-level pruning search space, the ablation study in Table 4 should include results for all single-level pruning strategies (not only block pruning, but also attention-head and linear pruning).
  - Knowledge distillation and adversarial distillation are also common approaches for obtaining lightweight diffusion models. Using part of the teacher model as a feature extractor for the discriminator is interesting, but a similar idea has already been explored in prior work [R2].
    - [R2] "SoundCTM: Unifying Score-based and Consistency Models for Full-band Text-to-Sound Generation," ICLR 2025.

- The main quantitative results shown in Table 2 lack several important metrics, such as dynamic degree and motion smoothness. For a comprehensive comparison, it is highly encouraged to provide the full list of results.

**Questions:**

- Could you list the empirical findings obtained through this work that are specifically important for text-to-video generation? It would be beneficial to demonstrate that this work offers sufficient novelty beyond simply combining existing techniques.

---

> ### Author Response · Authors · 2025-11-27
> **Author Response 1/2**
>
> **1. Increasing the compression ratio in VAE is a common choice (e.g., [R1]) for efficient latent diffusion models. To enhance the novelty, it would be beneficial if the authors could provide empirical findings that are particularly important in the context of video generation. [SANA]**
>
> We thank the reviewer for pointing this out. We fully agree that token reduction itself is a well-established technique for efficiency, widely adopted in standard Vision Transformers (e.g., DynamicViT) and recent image DiTs like SANA.
> We do not claim the concept of high compression VAE is novel, our contribution lies in the first systematic investigation of how these choices impact video generation, where the added temporal dimension creates unique stability and consistency trade-offs that prior works do not address. Our work provides specific empirical findings for on-device VAE design.
> 1. Sweet spot for mobile video generation **(Tab.3)**. We systematically benchmarked compression ratios from 4x16x16 to 8x64x64. We identified that moving from 4x16x16 to 8x32x32 yields a 20x speedup with negligible loss in VBench score. Crucially, we show that pushing further to 8x64x64, a common direction in aggressive image generation, causes a significant degradation in video generation.
> 2. Impact of latent channels. We explored the impact of latent channel width on diffusion performance. Contrary to the intuition that more channels preserve more information, we reveal that simply increasing channels (e.g. 128 → 512) improves VAE reconstruction PSNR but destabilizes the diffusion training, lowering the final VBench score.
> 3. Patchify vs. VAE compression **(Tab.10)**: We explicitly compared training DiTs on highly compressed VAE latents versus using lower-compression VAEs with larger patchify modules. Our results demonstrate that patch-2 achieves better performance in DiTs instead of patch-4 or higher compression ratios.
>
> **2. The idea of tri-level pruning sounds interesting, but the optimization process is greedy and not specifically designed for tri-level pruning. If the authors want to argue for the benefits of the tri-level pruning search space, the ablation study in Table 4 should include results for all single-level pruning strategies (not only block pruning, but also attention-head and linear pruning).**
>
> 1. The greedy approach is necessary because the combined search space is combinatorial and intractable. By dynamically selecting the action that minimizes the loss-to-parameter-reduction-ratio at each step, the algorithm automatically navigates the trade-off between the three levels.
> 2. We agree that single-level pruning comparisons are valuable, but they are structurally impractical in our specific scenario. Our goal is to compress a 2B parameter model to $\sim$950M which is less than 1B to satisfy mobile constraint (more than 50% reduction). As shown in the parameter breakdown of a standard DiT block below, the parameters are split roughly evenly between attention and FFN.
> 3. To achieve the target $\sim$950M budget using only Attention pruning, we would need to remove nearly  all Attention parameters. Similarly, using only FFN pruning would require removing the entire FFN. Both strategies would result in a broken model. Therefore, the only available single-level strategy for this compression rate is block pruning, which we’ve included and compared in Tab.5 in revision.
> 4. To further validate the benefit of the Tri-Level pruning, we conducted an additional experiment in the revision using a combined attention+FFN pruning strategy as shown in **Tab.5** narrow. We found the tri-level pruning consistently outperforms this dual-level strategy.
>
> |      Module     |                  Components                  | Parameter Count (approx.) |
> |:---------------:|:--------------------------------------------:|:-------------------------:|
> |  Self-Attention |                   QKV + Out                  |    $3d^2 + d^2 = 4d^2$    |
> | Cross-Attention |                   QKV + Out                  |    $3d^2 + d^2 = 4d^2$    |
> |       FFN       | Expansion ($d \rightarrow 4d \rightarrow d$) |           $8d^2$          |

---

> ### Author Response · Authors · 2025-11-27
> **Author Response 2/2**
>
> **3. Knowledge distillation and adversarial distillation are also common approaches for obtaining lightweight diffusion models. Using part of the teacher model as a feature extractor for the discriminator is interesting, but a similar idea has already been explored in prior work [R2 SoundCTM]**
>
> We fully acknowledge that step distillation and knowledge distillation (KD) are common techniques. However, they are necessary components specifically adapted to enable the first practical on-device DiT video generation.
> 1. Our knowledge distillation strategy is not a standalone addition but is highly incorporated with our tri-level pruning method. The tri-level pruning may create a student model with varying channel widths that do not match the teacher. To address this, we employ a specific feature alignment strategy using trainable affine transformations to map the pruned student’s features to the teacher’s space. Moreover, the alignment is applied group-wise to better align the teacher-student feature space instead of output only knowledge distillation.
> 2. Our step-distillation design appends learnable 3D self-attention and cross-attention layers (along with a classifier head) after the frozen blocks. This is critical for capturing long-range temporal dependencies and aligning the text condition with video, which prior feature extractors are not capable of. As shown in our ablation study in Tab.6, our design achieves a semantic score of 72.89 compared to only 67.78, 66.01 for a ResBlock-2D + Temporal attention or lightweight ResBlock design.
>
> **4. The main quantitative results shown in Table 2 lack several important metrics, such as dynamic degree and motion smoothness. For a comprehensive comparison, it is highly encouraged to provide the full list of results.**
>
> We thank the reviewer for this suggestion. We agree that dynamic degree and motion smoothness are critical metrics for evaluating video generation quality and we’ve included these two metrics in **Tab.2**. The full Vbench results are not included in the **Tab.2** due to space limitation.
> We’ve included the full VBench score of our model in **Tab.14, Appendix.L**.
>
> **5. Could you list the empirical findings obtained through this work that are specifically important for text-to-video generation? It would be beneficial to demonstrate that this work offers sufficient novelty beyond simply combining existing techniques.**
>
> We thank the reviewer for this question. Below we summarize the core empirical findings that are specifically important for text-to-video diffusion transformers:
> 1. VAE Compression Scaling Law for Mobile Video Diffusion. Moving from 4×16×16 to 8×32×32 yields 20× latency reduction with minimal VBench drop, while pushing further (e.g., 8×64×64) causes noticeable semantic and aesthetic degradation. This establishes a practical design rule: aggressively reduce spatial tokens to maximize until temporal fidelity begins to drop, and as of now 8x32x32 is a sweet point. **(Sec. 4.1, Tab. 3/4)**
> 2. Structural Sensitivity of Video DiTs. Through tri-level pruning and feature-aligned KD, we discover a consistent sensitivity hierarchy:
> **FFN width ≫ Block depth > Attention heads**.
> Which indicates that fine-grained reduction of heads has low impact and FFN compression causes the largest semantic/temporal collapse. From the design scope it is beneficial to keep wide FFNs but use less attention heads. This insight outperforms block-only pruning and width-only scaling **(Tab. 5, Fig. 3)**.
> 3. Discriminator Architecture Matters for Semantics. We show that standard lightweight 2D discriminators fail to capture spatiotemporal alignment. Our timestep-conditioned 3D-attention discriminator improves VBench-Semantic by +6.9 **(Tab. 6)**, which indicates that a video-aware discriminator is crucial for learning strong temporal dynamics in video diffusion step distillation.
> 4. On-Device Operator Bottleneck. FFN GEMMs are memory-bandwidth-bound on modern mobile SoCs. Our operator-level tiled GEMM resolves this hardware bottleneck, achieving 10%+ end-to-end speedup without kernel changes. **(Sec. 4.4, Fig. 6)**
>
> Together, these enable the first interactive-speed (15 FPS) DiT-based T2V generation on iPhone with strong VBench scores, a new proof-of-concept for large-scale generative AI running locally on users’ devices.

---

> > ### Comment · Reviewer_XXHh · 2025-11-28
> >
> > Thank you for your detailed response.
> >
> > While I appreciate the systematic investigation and engineering efforts, I still feel that the novelty in the methodology of this work is not particularly significant. In this context, the importance and generalizability of the experimental results become even more critical for the overall contribution. However, the current results seem to be highly specific to a particular target device, which makes it difficult to assess the broader impact and applicability of this study. I think it would be necessary to further discuss how generalizable the empirical insights obtained in this paper are beyond the specific device and setting.

---

### Official Review · Reviewer_YF34 · 2025-10-29

**Soundness:** 3
**Presentation:** 3
**Contribution:** 3
**Rating:** 8
**Confidence:** 3

**Summary:**

The paper proposes a series of novel optimizations to accelerate video generation and enable practical deployment on mobile
platforms. These optimizations allow the model to achieve approximately 15 frames per second (FPS) generation speed on an iPhone 16 Pro Max, demonstrating the feasibility of efficient, high-quality video generation on mobile devices.

**Strengths:**

1. The paper achieved the deployment of a video generation model on mobile devices (iPhone 16 Pro Max).
2. The experiments were extensive, involving extensive and complex work.
3. The final performance of the method is impressive.

**Weaknesses:**

1. It is recommended to verify the performance of the method on more mobile devices. There are currently various edge hardware environments; for example, low-power intelligent chips from Qualcomm and NVIDIA have been widely adopted, and they differ significantly from Apple's chip architectures. It is recommended to verify the performance on more types of chips or specify the hardware applicability scope of the method in the limitation section.
2. It is recommended to explore the causes and solutions for performance degradation induced by different VAEs, and strengthen the analysis of VAEs' impact on performance. For instance, it should be clarified which aspects of visual quality degradation correspond to PSNR loss, and how such degradation propagates to affect the sub-items in VBench scores. Additionally, efforts should be made to establish a theoretical connection between VAEs and DiT optimization, and to discuss the effectiveness and underlying principles of existing DiT optimization methods for compensating for performance losses when applied to models under current VAE conditions.
3. Given that this research has achieved significant improvements in video generation performance, it is recommended to make the code for reproducible experimental results publicly available to enhance the reproducibility of the paper's experimental results.

**Questions:**

The paper is generally clearly presented; for specific issues, please refer to the points listed in the Weaknesses.

---

> ### Author Response · Authors · 2025-11-27
> **Author Response**
>
> **1. It is recommended to verify the performance of the method on more mobile devices. ... Qualcomm and NVIDIA have been widely adopted, ... It is recommended to verify the performance on more types of chips or specify the hardware applicability scope of the method in the limitation section.**
>
> Thanks for the advice, we deploy our method on iPhone devices because the CoreML ecosystem enables a smooth speed benchmark and end-to-end demo, and this choice is consistent with popular practices in prior works such as MobileDiffusion, SnapFusion, and SnapGen-V. Other mobile platforms, such as Qualcomm-based devices, will require a startover in setup. We will add this hardware constraint to the limitations section, and leave the exploration of more mobile platforms as our future work. Here we demonstrate the speed of our model on older iPhone device to show generalization.
>
> |      Device      | Latency (ms) |
> |:----------------:|:------------:|
> | iPhone16 Pro Max |      668     |
> | iPhone15 Pro Max |      748     |
>
> **2. It is recommended to explore the causes and solutions for performance degradation induced by different VAEs, and strengthen the analysis of VAEs' impact on performance. For instance, it should be clarified which aspects of visual quality degradation correspond to PSNR loss, and how such degradation propagates to affect the sub-items in VBench scores. Additionally, efforts should be made to establish a theoretical connection between VAEs and DiT optimization, and to discuss the effectiveness and underlying principles of existing DiT optimization methods for compensating for performance losses when applied to models under current VAE conditions.**
>
> As compression increases from 4x16x16 (PSNR 33.1) to 8x64x64 (28.2), we notice that aesthetic falls from 64.45 to 55.29, semantic falls from 73.54 to 64.86, but temporal flickering and consistency remain similar (~98).
> The PSNR loss primarily represents the loss of high-frequency spatial details (texture, edges). This directly impacts the aesthetic score, which penalizes artifacts and the semantic score as the loss makes the object harder for the encoder to recognize.
> The temporal robustness indicates that the VAE bottleneck is largely spatial and the DiT successfully learns accurate motion dynamics even on highly compressed, lower-fidelity latents according to current benchmark metrics.
>
> **3. Given that this research has achieved significant improvements in video generation performance, it is recommended to make the code for reproducible experimental results publicly available to enhance the reproducibility of the paper's experimental results.**
>
> We thank the reviewer for this suggestion and fully recognize that reproducibility is essential for the community. While releasing full model weights is currently restricted due to policy, copyright and safety constraints, we provide several implementation components in the submission:
> 1. Hardware-aware pruning algorithm **(Algorithm 1 in Appendix C)**.
> 2. KD-guided pruning framework and sensitivity hierarchy **(Sec. 4.2, Fig. 3, Tab. 5)**.
> 3. Step-distillation architecture with DiT-specific discriminator design **(Sec. 4.3, Tab. 6)**.
> 4. VAE scaling and trade-off analysis with full metrics **(Sec. 4.1, Tab. 4/9/10)**.
> 5. Operator-level optimization details for tiled GEMM **(Sec. 4.4, Fig. 5/6)**.
>
> To further enhance reproducibility, we additionally release the following details as highlighted in the revision.
> 1. Detailed model configuration in **Appendix.D** (layer counts, FFN widths, attention heads, VAE choice).
> 2. Training schedules and hyperparameters  **Appendix.D** (teacher–student loss balancing, LR, warmup, sampling steps)
>
> These materials will allow future works to faithfully replicate the core training pipeline, model architectures, and on-device deployment performance reported in our paper.

---

### Official Review · Reviewer_Er4Q · 2025-10-29

**Soundness:** 3
**Presentation:** 3
**Contribution:** 2
**Rating:** 4
**Confidence:** 4

**Summary:**

The paper proposes a set of optimizations that make video Diffusion Transformers (DiTs) viable for a video generation on mobile devices. By combining a high-compression video VAE, KD-guided tri-level pruning, adversarial step distillation, and operator-level improvements via tiled GEMM, the authors achieve high-quality video synthesis at 15 FPS on an iPhone 16 Pro Max. The approach maintains strong visual fidelity while significantly reducing inference steps and model size, offering a practical solution for on-device video generation.

**Strengths:**

1) The paper is well written and motivated; it is easy to read and understand.
2) One of the pioneering works in the niche application: one of the first video diffusion transformer models running on-device with decent quality.
3) Comprehensive evaluation, both automated quality metrics and user studies are conducted, and good results are reported.
4) Somewhat novel design of adversarial step distillation setup adapted to the new architecture (DiTs).
5) Interesting discussions about quality-vs-efficiency trade-off in VAEs and pruning strategies of the original model.

**Weaknesses:**

1) My main concern with this work is limited novelty: while the engineering contributions are solid and well-executed, the paper primarily combines existing techniques, compression via VAE, pruning, distillation, and operator-level optimization, without introducing fundamentally new ideas or theoretical insights. The novelty lies more in integration than in conceptual advancement.
2) The pruning and distillation strategies, although tailored for DiTs, follow well-established paradigms. The tri-level pruning and adversarial step distillation are adaptations rather than breakthroughs, and similar approaches have been explored in UNet-based models and mobile optimization literature.
3) Another concern is about applicability of the proposed methodology to other mobile devices and platforms. Will the model perform as well on other IPhone devices for example? How about Android devices, will this approach extend to this platform in principle?

**Questions:**

1) Can the authors elaborate on what they consider the core novel contribution of this work beyond engineering integration? Specifically, how does the tri-level pruning or adversarial step distillation differ fundamentally from prior work in UNet-based or transformer-based models?
2) The paper focuses on text-to-video generation. Could the proposed pipeline be adapted for other tasks such as video editing, inpainting, or conditional generation (e.g., image-to-video)? If so, what modifications would be needed?

---

> ### Author Response · Authors · 2025-11-27
> **Author Response**
>
> **1. Limited novelty: while the engineering contributions are solid and well-executed, the paper primarily combines existing techniques, ..., than in conceptual advancement. ... Q1: Can the authors elaborate on what they consider the core novel contribution of this work beyond engineering integration? Specifically, how does the tri-level pruning or adversarial step distillation differ fundamentally from prior work in UNet-based or transformer-based models?**
>
> We thank the reviewer for the constructive feedback. We respectfully argue that our contributions go beyond the integration of existing techniques. We introduce a specific framework tailored for the unique constraints of video diffusion transformers, which differ fundamentally from prior UNet-based models, and more significant compared to image-to-text DiTs.
> 1. Tri-level pruning: Prior efficient DiT works, such as TinyFusion and SANA-1.5, primarily focus on block-level pruning (reducing network depth). We demonstrate that this coarse granularity is insufficient for mobile video generation. Our method extends the search space to a tri-level scheme (Blocks, Attention Heads, and FFN dimensions). Unlike generic pruning, our approach incorporates hardware capacity constraints (e.g., parameter budget <1B) directly into the search process. Guided by our sensitivity analysis **(Fig.3)**, this drives a novel pruning strategy—aggressively removing heads while conservatively pruning FFNs—that achieves a better trade-off than block pruning alone **(Tab.5)**.
> 2. Step-distill: While step distillation is essential for deployment , standard discriminator designs (e.g., lightweight ResBlocks or CNNs) struggle with the complex spatiotemporal dependencies of Video DiTs. We propose a new discriminator head design specifically for DiT. Instead of using external architectures, we repurpose the frozen pre-trained generator blocks as a timestep-conditioned feature extractor, appended with learnable 3D Self-Attention and Cross-Attention layers. Our ablation **(Tab.6)** confirms its superiority, showing it outperforms standard "ResBlock-2D + Temporal-Attn" and “Lightweight ResBlock” designs.
>
> **2. Another concern is about applicability of the proposed methodology to other mobile devices and platforms. Will the model perform as well on other iPhone devices for example? How about Android devices, will this approach extend to this platform in principle?**
>
> We thank the reviewer for this important question. We confirm that our proposed methodology is designed to be generalizable across different mobile platforms and generations, not limited to the iPhone 16 Pro Max.
> We evaluate our method on the iPhone by following prior works e.g., MobileDiffusion, SnapFusion, SnapGen-V.
> Our efficient architecture is not a static, hand-designed model but the output of an automated search algorithm. By adjusting a specific parameter budget and hardware constraint as inputs, it can be applied to other iPhone devices. While we showcased the iPhone 16 Pro Max to establish a state-of-the-art upper bound (15 FPS), the same search process can be executed with stricter constraints to generate optimal subnets for older devices, e.g., DiT latency is 748ms in iPhone 15 Pro Max vs. 668ms in iPhone 16 Pro Max. A key feature of our tiled GEMM optimization is its implementation at the operator level rather than the kernel level, which bypasses the limitation of closed-source compilers.
>
> |      Device      | Latency (ms) |
> |:----------------:|:------------:|
> | iPhone16 Pro Max |      668     |
> | iPhone15 Pro Max |      748     |
>
> **3. The paper focuses on text-to-video generation. Could the proposed pipeline be adapted for other tasks such as video editing, inpainting, or conditional generation (e.g., image-to-video)? If so, what modifications would be needed?**
>
> Yes, our proposed pipeline is highly adaptable to these tasks. Our framework targets the general DiT architectural and pipeline-level considerations of diffusion transformers rather than being tied to any specific task. Thus, our method can be straightforwardly extended to other downstream applications. For instance for the image-to-video task, we can keep the text embedding for our pruned 950M model but insert image condition through input padding and cross attention concatenation, similar to Wan-2.1, and then the same distillation pipeline can be applied.

---

### Official Review · Reviewer_U7Bc · 2025-11-01

**Soundness:** 3
**Presentation:** 3
**Contribution:** 3
**Rating:** 6
**Confidence:** 4

**Summary:**

This paper presents an efficient video generation framework that uses DiT to generate high-quality videos on mobile devices. The authors accelerate inference through techniques such as high-compression VAE, latency-aware pruning, adversarial step distillation, and GEMM-based tiling. Ultimately, the model achieves real-time video generation at 15 frames per second on the iPhone 16 Pro Max.

**Strengths:**

1.The authors propose an innovative model acceleration method that addresses the deployment challenges of DiT on mobile devices. By combining VAE compression, pruning, distillation, and other techniques, they achieve real-time video generation.

2.The experiments are thorough, with strong supporting arguments, and the writing is clear and easy to understand.

**Weaknesses:**

1.While the model performance is improved after three layers of pruning and distillation, the distillation process requires significant computational power, which makes training more challenging.

2.The framework was tested on the iPhone 16 Pro Max, but its performance may depend on the specific hardware architecture and optimization strategies. The differences in memory bandwidth and computational power across various edge devices could affect the model’s performance, especially on older devices.

**Questions:**

1.Regarding the memory bottleneck addressed by GEMM, is this a problem unique to mobile devices, or would using GEMM on a server-side system provide similar acceleration?

2.Distillation training requires substantial resources; how much time is required to train the lightweight model discussed in the paper?

3.During training, both synthetic and real data are used. Could you provide more details on the ratio between these two types of data, and how different ratios might affect the model’s performance?

---

> ### Author Response · Authors · 2025-11-27
> **Author Response**
>
> **1. The framework was tested on the iPhone 16 Pro Max, but its performance may depend on the specific hardware architecture and optimization strategies. The differences in memory bandwidth and computational power across various edge devices could affect the model’s performance, especially on older devices.**
>
> We thank the reviewer for this insightful comment. We agree that hardware capacity might differ in different devices. However, our pruning framework is explicitly designed to be hardware-aware, as shown in **Algorithm 1 (Appendix C)** and **Section 4.2.1**, the search process takes specific parameter budgets as inputs.
> Moreover, our use of Tiled GEMM addresses a universal bottleneck in mobile computing: limited memory bandwidth. We provide an exploration of this solution that is generalizable to other edge devices facing similar bandwidth constraints as shown **Fig.6** for memory bound operators. We show in the following table that our model can run on older devices with manageable cost, and benefits from more powerful devices.
>
> |      Device      | Latency (ms) |
> |:----------------:|:------------:|
> | iPhone16 Pro Max |      668     |
> | iPhone15 Pro Max |      748     |
>
> **2. Regarding the memory bottleneck addressed by GEMM, is this a problem unique to mobile devices, or would using GEMM on a server-side system provide similar acceleration?**
>
> The tiled GEMM strategy targets the bandwidth constraints specific to mobile SoCs, which currently still remain unsolved. Our analysis **(Fig.6)** confirms that large FFN layers are memory-bound on iPhone, where memory bandwidth is the bottleneck. In contrast, server GPUs (e.g., A100) possess typically sufficient HBM bandwidth, and existing libraries (cuBLAS) already handle kernel-level tiling. Therefore, our manual operator-level tiling serves as a necessary solution for edge devices, offering critical insights for future mobile compiler and kernel designs tailored to modern architectures like DiTs.
>
> **3. Distillation training requires substantial resources; how much time is required to train the lightweight model discussed in the paper?**
>
> While distillation requires extra training steps, it is a short fine-tuning phase (50K iterations) rather than a full pre-training cycle (typically 150K+ iterations). This allows the model to converge significantly faster and deliver better performance.
> In our ablation study **(Tab.5)**, the compact model without KD achieved only 79.00 Vbench score, compared to 81.45 for our KD-guided approach. This indicates that the extra computation power spent on distillation is necessary.
>
> **4. During training, both synthetic and real data are used. Could you provide more details on the ratio between these two types of data, and how different ratios might affect the model’s performance?**
>
> We only add a small amount of synthetic image (5%) for training. As mentioned in [1], [2] synthetic images are helpful in convergence speed. More synthetic data is helpful in convergence but may also harm generation quality due to the diversity limitation of synthetic data.
>
> [1] Zheng, Zangwei, et al. “Open-sora: Democratizing efficient video production for all” (2024)
>
> [2] Wan, Team, et al. "Wan: Open and advanced large-scale video generative models." arXiv preprint arXiv:2503.20314 (2025).

---

### Official Review · Reviewer_FawU · 2025-11-03

**Soundness:** 3
**Presentation:** 3
**Contribution:** 2
**Rating:** 4
**Confidence:** 3

**Summary:**

This paper presents a pipeline to run Diffusion Transformer-based video generation efficiently on mobile devices. It combines a high-compression video VAE, tri-level pruning with KD, 4-step adversarial distillation, and operator-level optimizations, achieving over 15 FPS video synthesis on an iPhone 16 Pro Max with competitive quality.

**Strengths:**

S1. This paper tackles a practical problem of on-device DiT video generation and provides good deployment results.

S2. The proposed pruning with a KD-guided framework yields substantial speedups with moderate quality drop.

S3. Demonstrating real-time performance on mobile hardware is a meaningful empirical result.

**Weaknesses:**

W1. The novelty is somewhat limited, as the proposed approach mainly combines existing compression, pruning, and distillation techniques, rather than introducing new algorithmic ideas.

W2. The contribution is engineering-driven, focusing on system and deployment optimizations, with relatively limited new ML insights or principles that generalize beyond this specific application.

W3. The evaluation is not fully convincing, as it lacks comparisons with recent efficient video diffusion and on-device methods, and provides limited analysis of efficiency-quality trade-offs.

**Questions:**

Q1. What is the key methodological novelty beyond adapting existing compression, pruning, and distillation techniques?

Q2. What ML insights or generalizable design principles does this work provide beyond the specific engineering optimizations for on-device DiT?

Q3. Can you add recent efficient diffusion and/or on-device baselines and deeper efficiency-quality trade-off analysis to better support the claims? Is it possible or not?

---

> ### Author Response · Authors · 2025-11-27
> **Author Response**
>
> **1. The novelty is somewhat limited, as the proposed approach mainly combines existing compression... What is the key methodological novelty beyond adapting existing compression, pruning, and distillation techniques?**
>
> We appreciate the reviewer’s point and clarify that our contributions go beyond a simple integration of existing components. Our key novelties include:
> 1. Holistic and hardware-aware co-design for video DiTs on edge devices. Unlike prior efficient image diffusion works, we target the video domain, guided by realistic hardware constraints. This enables a fundamentally different design space and trade-off analysis.
> 2. Systematic VAE compression design & analysis. We provide the first scaling study covering PSNR, VBench, and latency across compression ratios (4×16×16 → 8×64×64), revealing a sweet spot at 8×32×32. This balances token length (efficiency) and latent fidelity (quality). **(Tab. 3/4/9/10)**
> 3. Fine-grained KD-guided tri-level pruning. We jointly prune blocks, attention heads, FFN widths, guided by sensitivity analysis (Fig. 3), which reveals that FFNs are significantly more important than attention heads for video generation quality. This leads to >2× parameter reduction with only ~1.3% VBench drop **(Tab. 5)**. Prior video pruning works do not explore such structured, multi-level search.
> 4. Tiled GEMM for DiT inference on mobile hardware. We identify memory bandwidth, not compute, as the primary bottleneck for DiTs on mobile SoCs **(Fig. 6)**, and design a manual operator-level tiling, improving E2E FPS by 10%+ without kernel modification.
>
> In summary, our contribution is a unified optimization framework, introducing new empirical scaling laws, pruning insights, and operator-level deployment strategy, each specific to video DiTs on mobile.
>
> **2. The contribution is engineering-driven, ..., with relatively limited new ML insights, ..., Q2: What ML insights or generalizable design principles does this work provide beyond the specific engineering optimizations for on-device DiT?]**
>
> We respectfully disagree that our work is limited in ML insights. To the best of our knowledge, we provide the first systematic investigation in on-device efficient video generation using diffusion transformer at an interactive speed. We highlight the following ML insights provided by our study:
> 1. We provide an architecture level sensitivity hierarchy in DiTs. Contrary to prior pruning approaches, our sensitivity analysis **(Fig.3)** reveals a critical insight for DiT architectures in video generations. The FFN layer is the most critical factor for maintaining generation quality, whereas attention heads exhibit significant redundancy. Base on this, we proposed a more aggressive pruning on attention head/blocks while preserving more FFN dimensions. This finding provides a generalizable guideline for compressing diffusion transformer and model design.
> 2. We also provide the first systematic study on how VAE compression ratios contributes the DiT’s efficiency-quality trade-off. We identify a trade-off where aggressive compression (e.g., 8x64x64) degrades the generation quality, while low compression (e.g., 4x16x16) makes attention computationally expensive. Therefore, we proposed the 8x32x32 configuration as the sweet spot that balances token sequence length with generation quality of DiTs. This serves as a design guideline for future efficient video DiT design.
> 3. Patchify vs. VAE trade-offs. Patchification cannot replace VAE compression for DiTs due to degraded training stability **(Tab. 10)**.
>
> We believe these empirical findings provide new architectural guidelines for future efficient video DiTs.
>
> **3. The evaluation ... lacks comparisons with recent efficient video diffusion and on-device methods, and provides limited analysis of efficiency-quality trade-offs, ..., Q3: Can you add recent efficient diffusion and/or on-device baselines and deeper efficiency-quality trade-off analysis to better support the claims?**
>
> We do our best to compare existing works on efficient on-device video diffusion and the comparison is shown in **Appendix.B** and **Tab.8** (move to **Sec.5** and **Tab.3** in revision). Our method can provide consistent better VBench performance compared to prior works e.g., SnapGen-V and on-device Sora. Notably, our efficient DiT also delivers faster inference speed compared to on-device Sora. Efficiency–quality trade-offs are covered through: (i) VAE compression study (Tab. 3); (ii) Layer sensitivity analysis **(Fig. 3)**; and (iii) Step-distillation and pruning ablations **(Tab. 5/6)**.

---

### Author Response · Authors · 2025-11-27
**General Response**

We thank the reviewers for their thoughtful and constructive feedback.
We are glad that reviews recognize that our work tackles a practical problem, provides good empirical results, and delivers impressive performance.

As a general response, we also wish to highlight that our work presents a synergetic framework that brings together the first empirical study on scaling VAE compression and design; an analysis of DiT structural sensitivity along with a fine-grained pruning framework; a discriminator-head design specialized for DiTs; and operator-level optimizations addressing mobile memory bottlenecks.
Together, these advances enable the first to enable interactive-speed (15FPS) text-to-video generation on a mobile device, enlightening a promising research direction toward efficient and accessible video generation.

Regarding the concern raised by reviewers about evaluating our method only on Apple devices, we emphasize that the target device is used merely as a representative example for defining the search objective, and the proposed framework can be smoothly adapted to other hardware platforms. This practice is consistent with prior and recent work on efficient on-device diffusion models, such as MobileDiffusion [1], SnapFusion [2], SnapGen-V [3], and Neodragon [4], which either benchmark and design methods around specific target device [1,2,3] or propose techniques based on the characteristics of the device [4]. We believe our analysis and methodology address bottlenecks that are common across modern mobile SoCs, and our contributions are not tied to any particular device, making them broadly applicable across hardware. Furthermore, as detailed in our reply below, we additionally benchmark our approach on an older device (iPhone 15) to further support this claim.

We have addressed each question directly beneath the corresponding review and updated the manuscript accordingly, with all revisions clearly highlighted.
We appreciate your further comments and discussions!

[1] Zhao, Yang, et al. "Mobilediffusion: Instant text-to-image generation on mobile devices." European Conference on Computer Vision. Cham: Springer Nature Switzerland, 2024.

[2] Li, Yanyu, et al. "Snapfusion: Text-to-image diffusion model on mobile devices within two seconds." Advances in Neural Information Processing Systems 36 (2023): 20662-20678.

[3] Wu, Yushu, et al. "Snapgen-v: Generating a five-second video within five seconds on a mobile device." Proceedings of the Computer Vision and Pattern Recognition Conference. 2025.

[4] Karnewar, Animesh, et al. "Neodragon: Mobile Video Generation using Diffusion Transformer." arXiv preprint arXiv:2511.06055 (2025).

---

### Meta-Review · Area_Chair_bJxK · 2026-01-07

**Summary:**

The reviews agree the submission demonstrates an impressive engineering result: real-time (≈15 FPS) on-device text-to-video generation via a pipeline that combines VAE compression, structured pruning (with KD), step distillation, and operator-level optimizations. The main concerns driving the decision are (i) limited methodological novelty—per multiple reviewers, the work is largely an integration/adaptation of known techniques rather than introducing new algorithmic ideas or broadly applicable ML principles—and (ii) questions about generality and evaluation, including whether the empirical findings and optimizations transfer beyond the specific target device, and whether comparisons/trade-off analyses with recent efficient/on-device video diffusion baselines are sufficiently comprehensive.

Despite a compelling systems demonstration, multiple reviewers maintain that the work’s contribution is primarily engineering integration with uncertain generality beyond the specific device ecosystem, and at least one reviewer explicitly reiterates these issues after the rebuttal. The score distribution remains skewed toward borderline reject, making acceptance difficult to justify under typical ICLR novelty/generalizability expectations.

**Reviewer Concerns:**

Addressed concerns:
- The authors articulate novelty as (a) empirical scaling/design studies for VAE compression in video DiTs, (b) sensitivity-driven tri-level pruning behavior, (c) a DiT-specific discriminator-head design for step distillation, and (d) operator-level tiling motivated by mobile memory bottlenecks. This helps position the contribution, though it may not fully satisfy reviewers looking for fundamentally new ML methods.
- The rebuttal gives concrete responses on (i) why tiled GEMM is mobile-specific, (ii) the distillation schedule (50K iterations) and why it’s “fine-tuning,” and (iii) synthetic/real data ratio (reported as 5% synthetic). These questions appear substantively addressed.
- The authors claim they added dynamic degree and motion smoothness to the main table and provided full VBench results in an appendix. This addresses the request if the revision indeed includes them.

Remaining concerns:
- Even after the rebuttal, the core critique persists: the components are largely established, and novelty hinges on engineering integration + empirical observations.
- The authors add results on an older iPhone (iPhone 15) and argue portability in principle, but they do not provide evidence on non-Apple hardware (e.g., Android/Qualcomm/NVIDIA edge), and they acknowledge additional platforms would require substantial setup.
- The authors say they added comparisons (e.g., SnapGen-V / “on-device Sora”) and trade-off analyses via VAE scaling + ablations. This likely helps, but reviewers originally asked for “recent” baselines and deeper trade-off analysis; without seeing the revised breadth/strength of these comparisons, this remains a partial resolution.

**Reviewer Scores:**

Reviewer FawU: 4 → 4 or 5. Novelty concerns likely remain, but added comparisons and clearer framing could nudge to a weak borderline accept for a poster.

Reviewer U7Bc: 6 → 6 (possibly 7). Their questions were answered concretely; if the revision clearly incorporates the clarifications, they might increase slightly.

Reviewer Er4Q: 4 → 4 or 5. Rebuttal explains distinctions vs prior work and adds older-device results, but the “integration not conceptual advance” critique likely persists.

Reviewer YF34: 8 → 8. Already strongly positive; rebuttal aligns with their requested clarifications/limitations and reproducibility discussion.

Reviewer XXHh: 4 → 4. They explicitly follow up post-rebuttal maintaining limited novelty and emphasizing insufficient generalizability beyond the target device.

---

### Decision · Program_Chairs · 2026-01-26

Reject